# Large-scale characterisation of the nasal microbiome redefines *Staphylococcus aureus* colonisation status

*Staphylococcus aureus* colonises the nose in humans, with individuals defined as persistent, intermittent or non-carriers. Unlike the gut microbiome, the nasal microbiome has not been studied in large numbers of people. Here, we define the nasal microbiome in ~1100 individuals from the CARRIAGE study (ISRCTN: ISRCTN10474633) and combine with *S. aureus* culture data. We identify seven community state types (CST), including two CSTs more common in females. Approximately 70% of those who are persistently colonised with *S. aureus* have a CST dominated by *S. aureus*, while non-carriers are distributed across the other six CSTs. Intermittent carriers are not a unique state but have microbiomes that resemble non- or persistent carriers. Persistent carriage is positively associated with *S. aureus* abundance, and negatively associated with three *Corynebacterium* species, *Dolosigranulum pigrum*, *Staphylococcus epidermidis*, and *Moraxella catarrhalis*; the microbiome can be exploited with machine learning to accurately predict the persistence of *S. aureus* colonisation. Finally, we find that certain *S. aureus* lineages are better adapted to colonisation than others. Our data provides a comprehensive view of the nasal microbiome with respect to *S. aureus* colonisation, describing two key states: a *S. aureus* dominated CST in which *S. aureus* shapes the microbiome, and CSTs in which *S. aureus* is rare or absent.

The human nose is populated by a range of bacterial species which constitute the nasal microbiota, including the important commensal and opportunistic pathogen *Staphylococcus aureus*[1-3]. *S. aureus* nasal carriage is clinically important; carriers are at greater risk of *S. aureus* infection, often caused by the colonising strain[4-6] whilst decolonisation can reduce infection rates[7]. Based on longitudinal sampling, *S. aureus* nasal colonisation states have historically been divided into persistent, intermittent, and non-carrier[8]. However, it has been hypothesised that there may be only two biologically relevant categories (persistent carriers and non-carriers). Firstly, as compared to intermittent carriers and non-carriers, persistent carriers have higher *S. aureus* loads[9]. Secondly, after experimental colonisation, persistent carriers are more likely to select their autologous strain, become recolonised and maintain colonisation, whilst non- and intermittent carriers share similar *S. aureus* nasal elimination kinetics and anti-staphylococcal antibody profiles[10].

Multiple host factors have been identified that influence *S. aureus* carriage, and colonisation prevalence is higher in adult males and children[11-13], though many previous studies are based on small numbers of participants and in unrepresentative cohorts[12-14]. Interactions between the microbial residents of the anterior nares have been described between *S. aureus* and other members of the nasal microbiome[15-18]. For example, both *Staphylococcus epidermidis*[19] and *Staphylococcus lugdunensis*[20] have been shown to produce distinct compounds that inhibit *S. aureus* growth.

Unlike the gut microbiome, only a limited number of small studies have investigated the nasal microbiome. Yan et al. examined the nasal microbiome of 12 individuals at the anterior nares and two sites in the

✉ e-mail: d.aggarwal@imperial.ac.uk; eh6@sanger.ac.uk

inner part of the nasal cavity, revealing similarity in the dominant species between sites, but a lower overall diversity in the anterior nares compared to the middle meatus and spheno-ethmoidal recess, and amongst non-persistent carriers[3]. Analysis of the microbiome composition of culture-defined persistent or non-*S. aureus* carriers revealed that *S. aureus* had both an antagonistic relationship with *Corynebacterium pseudodiptheriticum* and synergism with *Corynebacterium accolens*, which was confirmed experimentally[3]. A larger study of eighty-six older (mean age ~65) twin pairs defined seven community state types using hierarchal clustering of Euclidean distance with Ward linkage (CST; distinct groups of bacteria) in the nose, identified negative associations of *S. aureus* with *Dolosigranulum sp.*, *Simonsiella sp.* and *Propionibacterium granulosum*, a positive association of *S. aureus* with *S. epidermidis*, and found a lower overall bacterial density as measured by 16S rRNA gene copy number and *S. aureus* 16S gene copy abundance amongst women[17]. In early life, the assembly of the nasal microbiome is weakly influenced by the maternal microbiome, while environmental exposures such as daycare have a greater impact[18; furthermore], both the mode of delivery and breast feeding have been found to influence the infant nasal microbiome with some variation between studies (reviewed in[21,22]). It is clear that *S. aureus* colonises the nasopharynx in the first weeks of life[23], and several species have been reported to support *S. aureus* colonisation in infants[18]. This colonisation is in turn inversely correlated with maternal *Dolosigranulum pigrum*[18]. More recent meta-analysis of paediatric (*n* = 99 individuals) and adult (*n* = 210) 16s rRNA gene-based studies revealed co-occurrence within the broader microbial community between *D. pigrum* and both *C. pseudodiptheriticum*, and *Moraxella nonliquefaciens* in children[16]. This trinity of species have been reported to be associated with greater stability of the nasal microbiome in early life[24]. Similarly, in adults, four different *Corynebacterium* species have been found to be positively associated with *D. pigrum*, which in turn was negatively associated with *S. aureus*, an association that was further demonstrated in vitro[16].

The use of antagonistic bacterial strains as live biotherapeutics (probiotics) is an attractive option to reduce *S. aureus* nasal colonisation without the need for antibiotics. This concept was demonstrated using a *Corynebacterium* sp. to successfully eradicate *S. aureus*[25]. In more recent work, *S. aureus* was reported to be excluded from the gut in the presence of *B. subtilis* via inhibition of pathogen signalling[26]; this was translated into a clinical trial of *B. subtilis* as a live biotherapeutic, which was successful in eliminating viable *S. aureus* from the gut and reducing, but not eradicating, the bacterial loads in the nose[27].

In summary, while interactions between bacterial species in the nasal microbiota have been identified, including with *S. aureus*, nasal microbiome studies have only involved small sample sizes and used selected populations, which likely reduces the generalisability of the findings. In addition, in recent years the importance of systematic removal of contamination in microbiome studies, particularly lower biomass/complexity environments such the nasal microbiome, has been established[28–30]. This means studies (particularly those with small sample sizes) that have not systematically removed contamination risk being confounded. Critically, to understand the nasal microbiome in relation to *S. aureus* colonisation, no microbiome study of greater than forty individuals[31] has included *S. aureus* colonisation status as defined by longitudinal sampling and culture, which has been used to understand *S. aureus* colonisation for 70 years[32].

Here, we utilise microbiome data from nasal swabs of 1180 generally healthy community participants from across England in the CARRIAGE study, along with three weekly nasal swabs cultured for *S. aureus* to determine the microbiome structure associated with nasal *S. aureus* carriage, including evaluation of the validity of the current defined *S. aureus* colonisation states (persistent, intermittent and non-carriers). We show that persistent *S. aureus* carriage is strongly associated with a distinct nasal microbiome CST dominated by *S. aureus*, while non-carriers exhibit diverse CSTs with low *S. aureus* abundance; intermittent carriers are not a unique state but have microbiomes that resemble non- or persistent carriers. We show that machine learning models leveraging microbiome composition can accurately predict colonisation persistence, and that certain *S. aureus* lineages are more adept at establishing nasal colonisation.

## Results

### Determination of the nasal microbiome in a large cohort

To study the biological basis of *S. aureus* colonisation in this observational cohort study, samples were taken from the anterior nares from generally healthy human volunteers from the community participating in the CARRIAGE study[33] between 13th October 2016 and 17th May 2017 from across England. *S. aureus* colonisation status was assessed by culture of three self-administered nasal swabs delivered to participants and taken at weekly intervals, and subsequently posted back to the laboratory (Fig. 1a). *S. aureus* colonisation status was defined as: (i) persistent colonisation, 306/1091 (28.0%), based on three *S. aureus* culture positive weekly nasal swabs, (ii) intermittent colonisation, 191/1091 (17.5%), defined as one or two swabs positive, and (iii) non-carrier status, 594/1091 (54.4%), defined as no swabs positive, based on previous studies[9,10,34,35] (89 failed to return all samples). Lifestyle information was collected by questionnaires or from pre-existing data held as part of baseline questionnaires in previous studies involving the same participants. The Amies transport liquid that the swabs (the same swabs that were used for culture) were transported to the laboratory in were processed without culture for 16S rRNA gene sequencing to identify the microbial community composition (Supplementary Fig. S1 and 2). Participants had a mean age of 51.4 (median, 53) and 52.8% were female. A total of 1756 samples, which included the first swabs of 1180 participants underwent 16S rRNA gene sequencing to determine the microbiome composition (Supplementary Fig. S1). After quality control (QC) (see Methods and Supplementary Table S2), 1055 samples remained, and after rarefaction and a systematic analysis to remove any likely contaminants, 53 Operational Taxonomic Units (OTUs) (24 species level taxa) remained.

### Differences in within-sample microbiome diversity

We first investigated variation in Alpha diversity (measures of within-sample diversity) by culture-defined *S. aureus* colonisation status, to determine differences in the microbiome between *S. aureus* colonisation states. We found Alpha diversity was significantly lower in samples from persistent carriers when compared to non-carriers or intermittent carriers when using either the Shannon or Simpsons diversity metrics (both $p < 0.001$), and found no significant difference between non-carriers and intermittent carriers using either (both $p > 0.2$) (Fig. 1b, c).

We next investigated Beta diversity (similarity or dissimilarity between two samples) (Fig. 1f–i) using Bray-curtis distance by colonisation status, which differed significantly (PERMANOVA analysis, $F(2) = 36.67$, $p < 0.001$). Distinct separation of samples from persistent- and non-carriers could be observed by non-metric multidimensional scaling (NMDS) (Fig. 1f) and PCoA (Fig. 1g) ordination plots; Beta dispersion (PERMDISP) analysis showed significantly greater within-group variability in persistent carriers compared to intermittent carriers and non-carriers ($p < 0.001$ for both) and such variance differences may contribute in part to the PERMANOVA result. However, samples from intermittent carriers did not form a distinct cluster, and instead overlapped within the persistent or the non-carrier clusters, but with more samples from intermittent carriers being clustered with the non-carriers as visualised by the overlap in data ellipses in Fig. 1f, g. This suggests that the microbiomes of intermittent (or rather occasionally *S. aureus* culture-positive) carriers are not distinct but typically more similar to non-carriers, with smaller

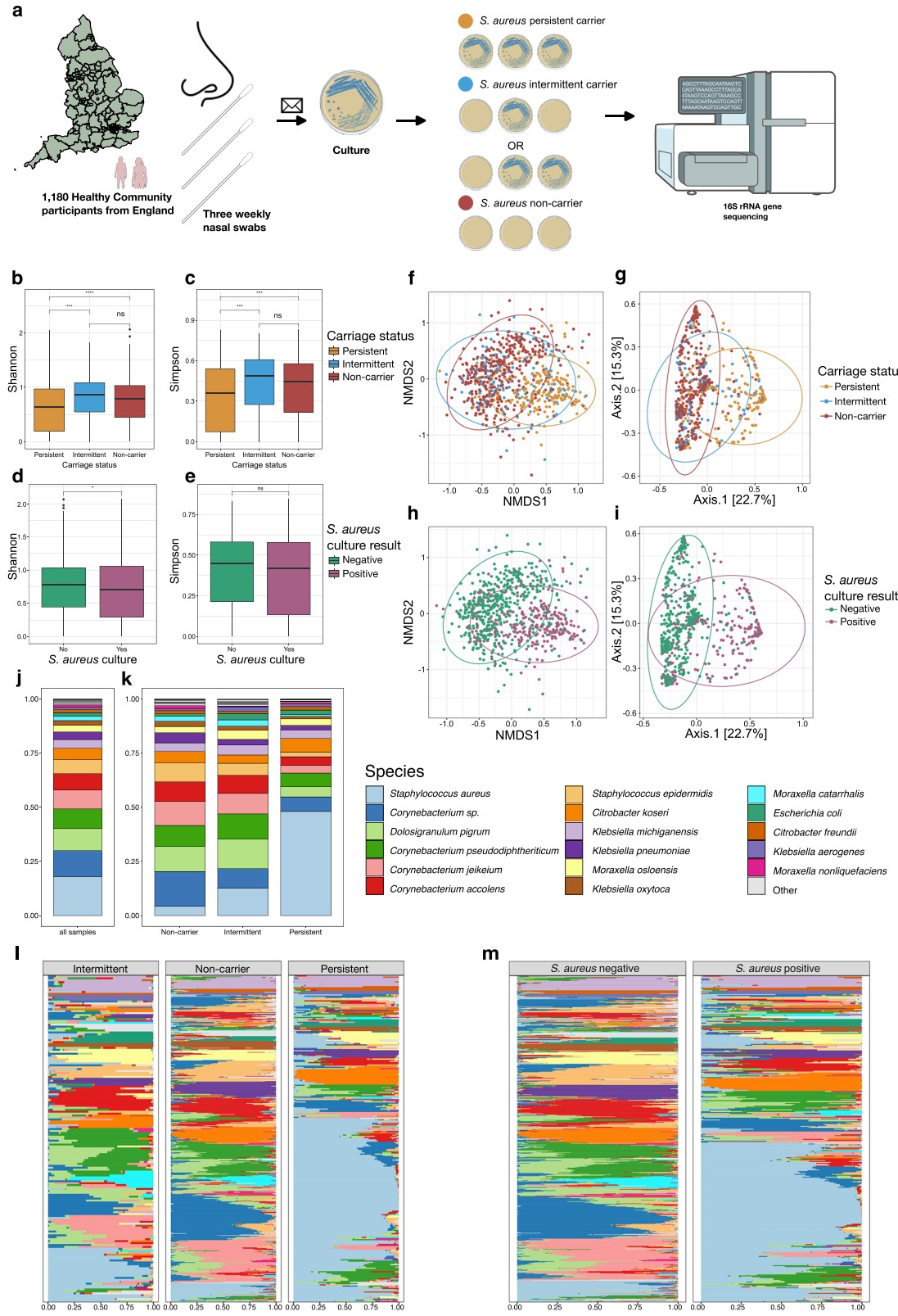

numbers that have similar microbiomes to persistent carriers. Likewise, we observed similar distinct clusters between *S. aureus* culture-positive and culture-negative samples on the ordination plots (Fig. 1h–i). Again, the two groups defined by *S. aureus* culture result differed significantly by PERMANOVA analysis ($F(2) = 59.01$, $p < 0.001$) (Fig. 1h, i). We only observed an

association of sex with variation in the Bray-Curtis distance, given females (115/543, 21.2%) are less commonly persistent carriers compared to males (160/511, 31.3%) ($p < 0.001$), but with a low F statistic and $R^2$ values ($F(2) = 2.83$, $p = 0.006$, $R^2 = 0.39\%$). There was no association with smoking, pet ownership, healthcare worker, chronic skin condition, and diabetes.

**Fig. 1 | Study design and nasal diversity and composition by _Staphylococcus aureus_ colonisation status and _Staphylococcus aureus_ culture result.**
**a** illustration of study design and cohort created in BioRender. Ng, D. (2025) https://BioRender.com/r99ahyk. **b–e** Box plots comparing Alpha diversity (Shannon and Simpson) from nasal samples by (b,c) _Staphylococcus aureus_ colonisation status: persistent ($n = 210$), intermittent ($n = 120$), and non-carriers ($n = 413$) and **d**, **e** _Staphylococcus aureus_ culture result: negative ($n = 510$), positive ($n = 284$). Each data point is derived from a nasal sample from a distinct individual. The midline of the boxplot represents the median value; the lower limit of the box represents the first quartile (25th percentile), and the upper limit of the box represents the third quartile (75th percentile); the whiskers (upper and lower) extend to the largest and smallest value from the box, no further than 1.5*IQR from the box. Asterisks indicate statistical significance from pairwise comparisons using the Wilcoxon rank-sum test (two-sided). Significance levels are denoted as follows: not significant (ns), $p \leq 0.05$ (*), $p \leq 0.01$ (**), $p \leq 0.001$ (***), and $p \leq 0.0001$ (****). **f–i** Ordination plots representing Beta diversity by Bray-Curtis distance and coloured by **f**, **g** _Staphylococcus aureus_ colonisation status and **h**, **i** _Staphylococcus aureus_ culture result. **a–i** _Staphylococcus aureus_ colonisation status colours: persistent, orange; intermittent, blue; non-carrier, red. _Staphylococcus aureus_ culture result colours: negative culture, green; positive culture, purple. **f**, **h** show NMDS plots **g**, **i** show PCoA plots. **f**, **g** Bray-Curtis distance between colonisation states significantly

differed by one-way PERMANOVA analysis ($F(2) = 36.67, p < 0.001$). Beta dispersion (PERMDISP) analysis showed significantly greater within-group variability in persistent carriers compared to intermittent carriers and non-carriers ($p < 0.001$ for both). No difference was observed between intermittent carriers and non-carriers ($p = 0.99$). Given the ordination plots, the observed differences in beta diversity appear to reflect both shifts in community composition and variation in dispersion. **h**, **i** Bray-Curtis distance between _Staphylococcus aureus_ culture positive and negative samples significantly differed by one-way PERMANOVA analysis ($F(2) = 59.01, p < 0.001$). Beta dispersion (PERMDISP) analysis revealed significantly greater within-group variability among culture-positive individuals (F = 47.2, $p = 0.001$), suggesting potential heterogeneity in dispersion. Again, the ordination plots showed distinct clustering by group, supporting a shift in community structure rather than an artefact of dispersion. Data ellipses represent the 95% confidence level that values lie within this space, assuming a multivariate t-distribution. **j** Mean microbial composition of all samples at a species level **k** Mean microbial composition of samples by _S. aureus_ colonisation status at a species level. **l** Microbial composition of samples across the study dataset, separated by colonisation status at a species level **m** Microbial composition of samples across the study dataset, separated by _S. aureus_ nasal swab culture result at a species level. **j–m** Top 17 species represented. **l**, **m** Samples sorted by Bray-Curtis similarity.

## Compositional differences by colonisation status and defining community state types

To visualise the causes of differences observed in Alpha and Beta diversity, we analysed species composition by _S. aureus_ colonisation status (Fig. 1j, k). The lower Alpha diversity of the persistent carriers was associated with the dominance of _S. aureus_ in the species composition of this groups, compared to the intermittent and non-carriers. In contrast, the nasal microbiome of non-carriers is largely dominated by multiple _Corynebacterium_ species and _D. pigrum_. We next examined species composition at the level of each participant's sample, separated by colonisation state (Fig. 1l). This showed that amongst the 275 persistently colonised participants, _S. aureus_ was the dominant organism ( > 50% of reads) for 136/275 (49.5%), and in a subset of 96/275 (34.9%) participants, _S. aureus_ represented >75% of reads. In comparison, >50% of reads from _S. aureus_ was only seen in the 22/532 (4.1%), of _S. aureus_ culture-negative (non-carriers) and 26/169 (15.4%) occasionally _S. aureus_ culture-positive individuals (intermittent carriers). Instead, the non-carriers and subset of intermittent carriers were clearly dominated by three different _Corynebacterium_ species (_C. pseudodiphtheriticum, C. jeikeium, and C. accolens_) at abundances not seen in _S. aureus_ persistent carriers (Fig. 1l).

Classifying individual samples by _S. aureus_ culture result revealed that _S. aureus_ was the predominant species ( > 50% of reads) in 164/382 (42.9%) of the _S. aureus_ culture-positive samples, and only 32/672 (4.76%) of culture-negative samples (Fig. 1m). Men are known to have higher _S. aureus_ culture positive rates[36,37] and here we found 213/511 (41.7%) swabs returned from male participants were positive for _S. aureus_ on culture compared to 169/543 (31.1%) swabs from female participants. Given our finding that a small proportion of _S. aureus_ culture negative samples have a high _S. aureus_ abundance, we examined the possibility of bias in _S. aureus_ culture by sex; however, we did not find that culture-negative samples with a higher _S. aureus_ abundance ( > 50% of reads) were more prevalent amongst females (17/32, 53.1%) compared to males (15/32, 46.9%). Expectedly, low _S. aureus_ abundance was associated with a _S. aureus_ culture negative result; 92/672 (13.7%) culture negative samples contained no _S. aureus_ reads, whilst 550/672 (81.8%) culture negative samples contained <1% of _S. aureus_ reads (Supplementary Fig. S6).

## Community state types

Next, we generated a heatmap of taxa abundance (Fig. 2a), organised by hierarchical clustering by Bray-Curtis distance to examine the relationships between microbial residents of the anterior nares. We

used this to define community state types (CSTs), i.e. samples with similar abundances of species which cluster together. To determine the number of clusters in the data, we calculated a gap statistic with ordination values using Bray-Curtis distances (Supplementary Fig. S7). A total of 7 clusters were defined; we identified CSTs from the heatmap plot (Fig. 2a). CST VII, representing a diverse group of sub-clusters is further detailed in Supplementary Fig. S8. From the heatmap, it is evident that individuals always _S. aureus_ culture-positive (persistent carriers) cluster to form the majority of CST I (72.4%, 155/214), whilst those always _S. aureus_ culture-negative (non-carriers) are represented largely by the remaining CSTs (Fig. 2b, c). Intermittent carriers are dispersed across the CSTs. Using a multinomial logistic regression model, we found men had a reduced relative risk for association with CST VI (OR = 0.53, 95% CI = 0.30–0.93, $p = 0.03$) and CST VII (OR = 0.67, 95% CI = 0.47–0.96, $p = 0.03$) compared with CST I (Fig. 2e). No other significant associations with CSTs were observed. Adjusted odd-ratios are provided in Supplementary Table S5.

We then formally evaluated differences in species abundances by colonisation status using ANCOM-BC2, which minimises the false discovery rate, using the unadjusted read count table. When comparing always _S. aureus_ culture-negative individuals (non-carriers) and always culture-positive individuals (persistent carriers) carriers, a significant positive association of _S. aureus_ was seen with persistent carriage (as expected), and a significant negative association was seen with multiple _Corynebacterium_ species, _D. pigrum, S. epidermidis_, and _M. catarrhalis_ (Fig. 3a and Supplementary Table S4). No significant differences in species abundance other than _S. aureus_ between non-carriers and intermittent carriers was observed (Fig. 3a and Supplementary Table S4). Notably, persistent carriers had a greater log-fold change in _S. aureus_ when compared with occasionally _S. aureus_ positive individuals (intermittent carriers), in comparison to non-carriers, suggesting the relative abundance of _S. aureus_ may be driving its longitudinal carriage.

To further explore the interactions between the different members of the nasal microbiome we generated a co-occurrence network (Fig. 3b). This comprised 24 taxa and 98 edges (density = 0.18; average degree = 8.17; clustering coefficient = 0.43). Several taxa including _D. pigrum, S. aureus_, and _C. pseudodiphtheriticum_, exhibited high degree centrality (a measure of how many direct connections each taxon has). The network resolved into four subcommunities (Q = 0.44). Hub taxa identified by eigenvector (EV) centrality (which weights the number of connections and the importance of connected neighbours; Supplementary Table S6) included _D. pigrum_ (Group 1, EV = 1.00), _S. aureus_

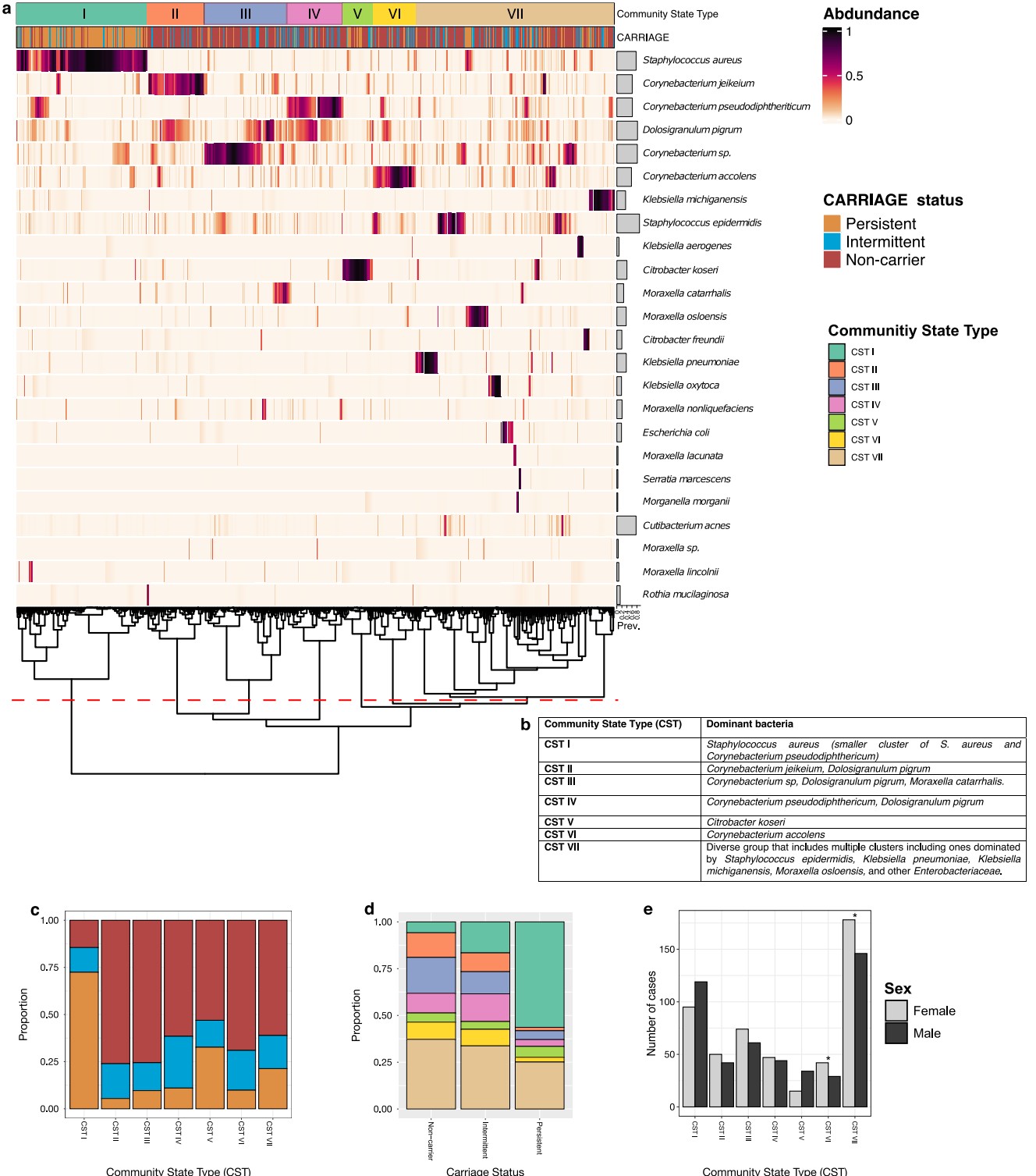

**Fig. 2 | Microbial community state types observed in the anterior nares.**
**a** Heatmap of species abundances in CARRIAGE nasal samples. Samples are ordered by hierarchical clustering using Bray-Curtis distances based on the compositional, relative abundance data, represented by the dendrogram. Prevalence of each species across the samples is represented by the horizontal bar plots. Community state types (CSTs) and *S. aureus* colonisation status of samples are represented above the heatmap. Seven distinct CSTs were identified from the selection of hierarchical clusters determined by calculating a gap statistic on the Bray-Curtis

distance. **b** Bacterial species dominating each CST. **c** Composition of CSTs by colonisation status (persistent, orange; intermittent, blue; non-carrier, red). **d** Composition of *S. aureus* colonisation status by CSTs. **e** Composition of each CST by sex. Using a multinomial logistic regression model, we found men had a reduced relative risk for association with CST VI (OR = 0.53, 95% CI = 0.30-0.93, *p* = 0.03) and CST VII (OR = 0.67, 95% CI = 0.47-0.96, *p* = 0.03) compared with CST I. *significant difference (*p* < 0.05).

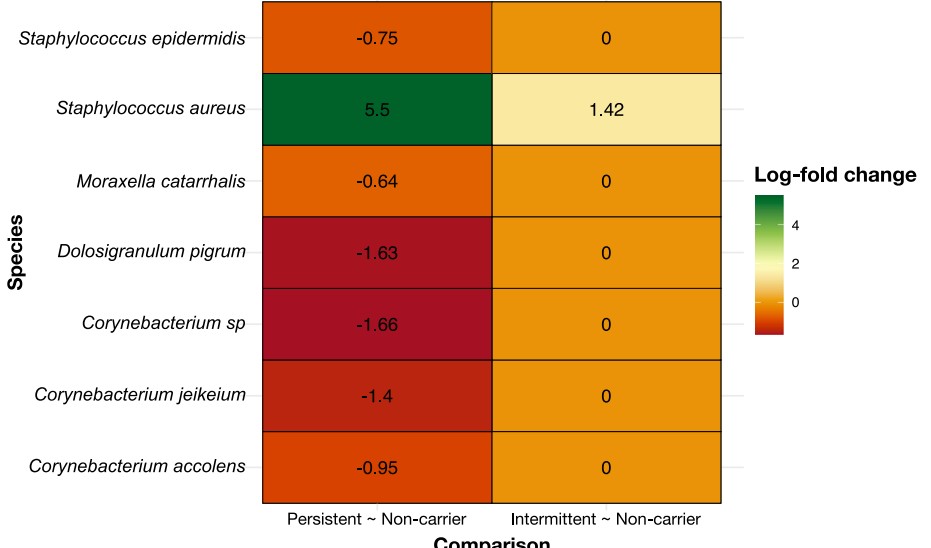

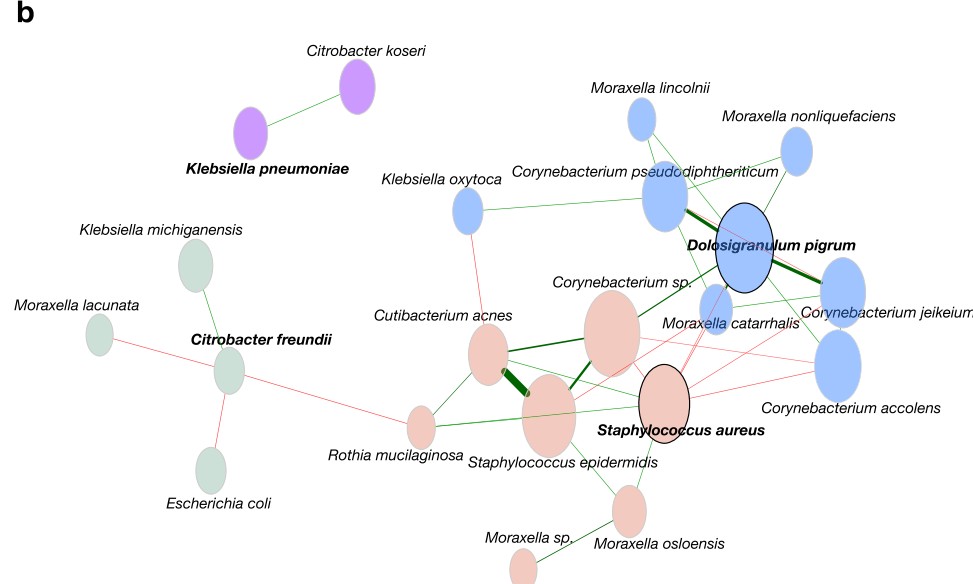

**Fig. 3 | Differential abundance, stratified by Staphylococcus aureus carriage status, and co-occurrence network structure of species observed in the anterior nares. a** Differential abundance of species by nasal colonisation status using ANCOM-BC2. Log-fold (natural log) changes as compared to *S. aureus* non-carriers. Column one compares persistent carriers against non-carriers and column two compares intermittent carriers against non-carriers. **b** Species-level networks inferred with NetCoMi (v1.2)[70] using SparCC correlations (zeroes replaced by a pseudocount; centred log-ratio (CLR) transform; 1000 bootstraps). Network presented with spring layout, plotted with nodes coloured by group and sized by CLR abundance. Species in bold represent the hub taxa for each group.

(Group 2, EV = 0.79), *Citrobacter freundii* (Group 3, EV = 0.07), and *Klebsiella pneumoniae* (Group 4, EV = 9.88×10⁻¹⁷). These may represent keystone roles in community organisation.

We next examined the stability of the community in the anterior nares, in a subgroup of 34 participants, from the rarefied dataset to 10,000 reads, two or three samples (*n* = 75) were available over consecutive weeks (Supplementary Fig. S9-12). These included 13 persistent carriers, 7 intermittent carriers, and 14 non-carriers. We correlated pairwise Alpha diversity of participants (i.e. comparison of diversity indices from samples of the same participant between consecutive weeks) by colonisation status. Persistent (Spearman's rho = 0.54, *p* = 0.028) and intermittent carriers (Spearman's rho = 0.79, *p* = 0.028) were found to have greater stability compared to non-carriers (Spearman's rho = 0.30, *p* = 0.268).

**Further examining the microbiome of 'intermittent' carriage**

Having observed that the majority of microbiomes of intermittent carriers clustered with those of the *S. aureus* non-carriers group (e.g. overlapping data ellipses in Fig. 1f, g), we hypothesised that intermittent carriers could be misclassified non- or persistent carriers. We examined differences in Alpha diversity between the one and two swab positive intermittent subgroup (Supplementary Fig. S13), using only samples with greater than 10,000 reads. We found no significant difference in Alpha diversity when comparing samples with one *S. aureus* positive swab compared with two (*p* = 0.21). Beta diversity by Bray-Curtis index between samples with one or two positive *S. aureus* swabs did differ significantly by PERMANOVA analysis (*F*(2) = 3.19, *p* = 0.003), suggesting that these groups have differing microbial compositions (Fig. 4a–d).

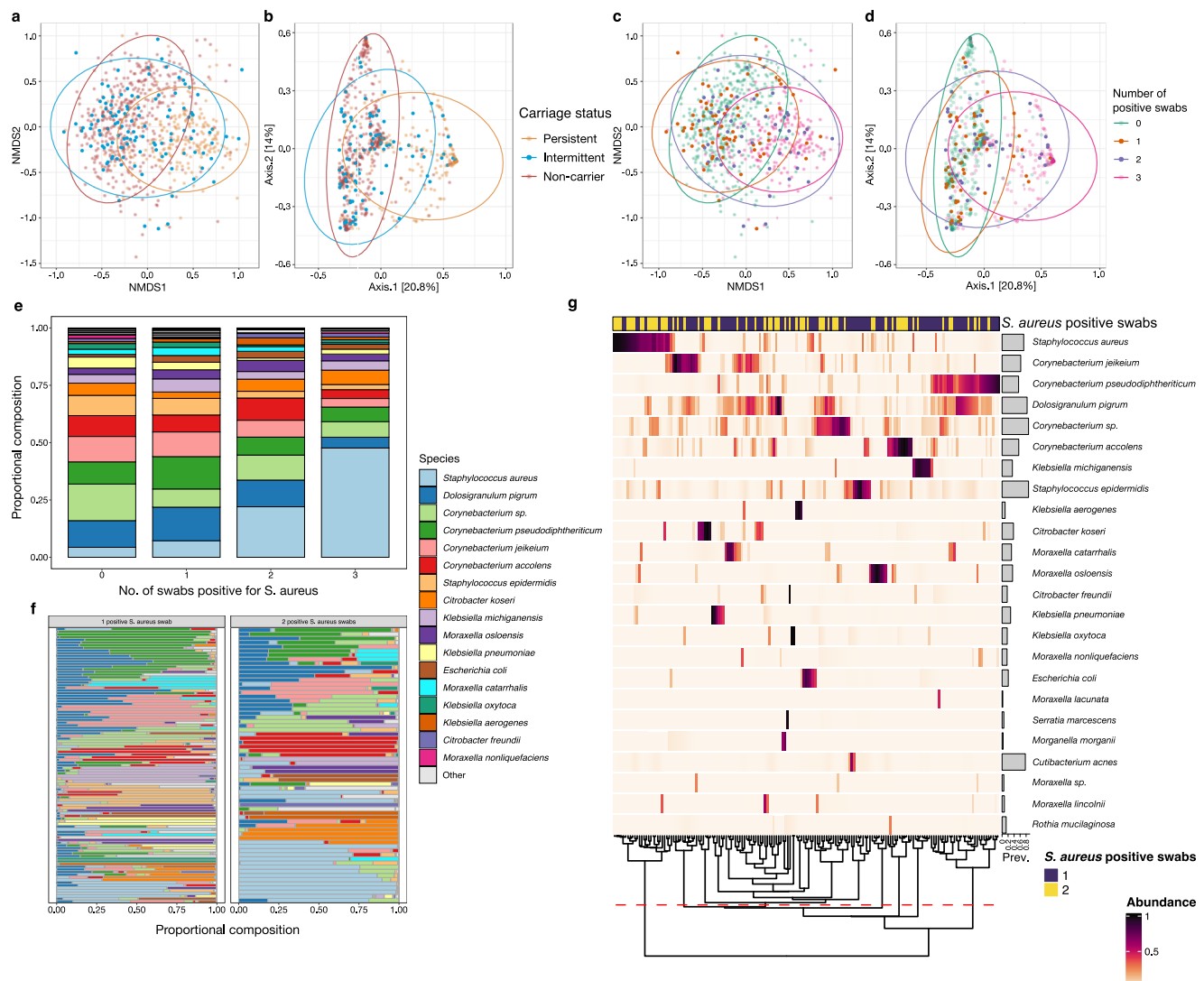

**Fig. 4 | Microbial composition of species in the anterior nares by the number of positive *S. aureus* swabs, with a focus on intermittent carriers. a–d** Ordination plots representing Beta diversity by Bray-Curtis distance and coloured by *Staphylococcus aureus* colonisation status and the number of *S. aureus* culture-positive swabs relating to each participant represented. **a, c** show NMDS plots **b, d** show PCoA plots. **a, b** Fig. 1 panels f to g have been reproduced to highlight the distribution of intermittent carriers (blue) and with *S. aureus* non-carriers (red) and persistent carriers (blue) faded into the background. The values representing intermittent carriers on the ordination plots of Bray-Curtis distance visibly span both the non-carrier and persistent carrier clusters. **c, d** These plots represent the same Bray-Curtis distances as shown on panel a to b but with points coloured by the number of positive swabs from the participant. Despite limited numbers, it is apparent that there is greater overlap of the non-carriers (0 positive swabs, green) with the participants with 1 positive swab individuals (orange), and a similar relationship is seen between the participants with 2 positive swabs (purple) and the

persistent carriers (3 positive swabs, pink). **e** Mean abundance by the number of swabs positive for *S. aureus* including non-, intermittent and persistent carriers. **f** Microbial composition represented by relative abundance of species residing in the anterior nares of individual intermittent carriers, comparing the number of *Staphylococcus aureus* culture positive swabs obtained (one vs two). **g** Microbial composition of the anterior nares from intermittent carriers represented as a heatmap. The number of samples positive for *S. aureus* (1, purple; 2, yellow) from the participant associated with the represented participant sample is shown in the bar above the heatmap. Samples are ordered by hierarchical clustering using Bray-Curtis distances on the compositional relative abundance data. Prevalence of each species is highlighted in the horizontal bar plots. The dashed red line represents splitting of hierarchical clustering dendrogram in seven community state types, as determined by the gap statistic. Participants with two positive swabs appear to have a higher abundance of *S. aureus*.

We next explored the abundance of species across the samples depending on the number of swabs which were positive for *S. aureus*. There is a clear continuous trend in the variation in abundance from zero to three positive swabs (Fig. 4e). We then subset the participants representing intermittent carriers (n = 169) from the dataset to examine if these were two distinct populations (rather than one) based on the number of *S. aureus* positive swabs (one swab, n = 103 and two swabs, n = 66). From examination of the species composition of individual samples, a different microbial community structure is apparent

for intermittent carriers who are positive for two swabs compared to those with one swab (Fig. 4f).

We formally analysed the differences in community structure using a heatmap of abundances from the samples of intermittent carriers, which displays a similar structure of clustering to that observed when comparing persistent carriers (Fig. 4g). Again, we calculated a gap statistic, giving an optimal number of CSTs of 7 (same as full dataset), and the hierarchical clustered dendrogram was split accordingly (Fig. 4g). On this heatmap, it is clear that the CSTs that are

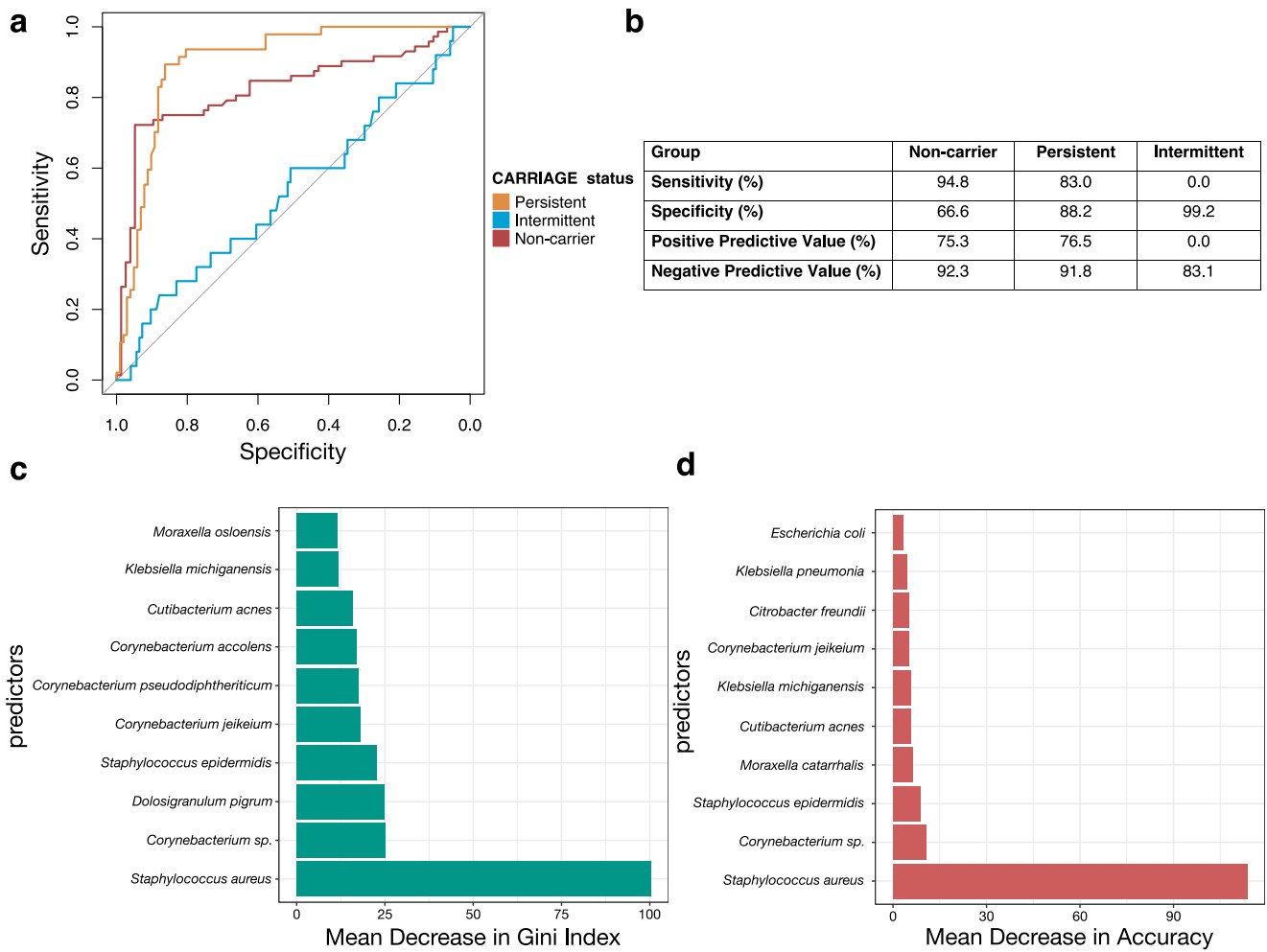

**Fig. 5 | Random forest classifier of the nasal microbiome data. a** ROC curves demonstrating model performance for classification of non-carriers vs others (grey line), persistent carriers vs others (blue line), and intermittent carriers vs others (red line). The multi-class area under the curve was calculated as 76.8%. **b** Performance of the random forest model to predict the nasal microbiome. Values provided as percentages. **c** Feature importance as determined by mean decrease in gini index from the random forest classifier. **d** Feature importance as determined by mean decrease in model accuracy from the random forest classifier.

dominated by *Corynebacterium* species, *D. pigrum* and *S. epidermidis* are associated with samples where participants had one positive *S. aureus* culture. 18/66 (27.3%) intermittent carriers who had two positive *S. aureus* cultures were associated with a CST dominated by *S. aureus*, compared to 7/103 (6.8%) with one positive swab. These findings reflect similar observations for persistent carriers and non-carriers, respectively (Fig. 1F, G), and provide further evidence that, with respect to the underlying microbiome, intermittent carriers do not possess a distinct phenotype and are either similar to persistent carriers or non-carriers.

**Predicting colonisation status from the nasal microbiome**

We next used a random forest model to establish whether microbiome data could be used to predict the culture-based categorisation of nasal *S. aureus* colonisation status. Additionally, this served as a sensitivity analysis for the previous differential abundance analysis (Fig. 3A), which allows for the identification of significant microbial determinants for *S. aureus* colonisation status. We split the data into training and test data at a ratio of 80:20, and determined the best number of candidates to be sampled at each tree (mtry) to be 6. The estimated test classification accuracy of the trained model was 73.2% (1-estimated out of box error) with the lowest class error for non-carriers (6.85%) and highest for intermittent carriers (100%).

We determined the accuracy, sensitivity and specificity of the model with the test data. The overall accuracy of the model was 75.2% (95% CI = 67.4%-81.9%, *p* < 0.001) significantly exceeding the no information rate (Fig. 5A). Overall, the model performed best in predicting persistent colonisation with 83.0% and 88.2% sensitivity and specificity, respectively, suggesting the greatest utility for identification of individuals at higher risk of persistent *S. aureus* colonisation (Fig. 5B). For non-carriers, the sensitivity was higher at 94.8%, but specificity lower at 66.6%. For intermittent carriers the sensitivity was 0.0% suggesting the model was completely unable to predict the intermittent colonisation from the microbiome data; of the 25 intermittent carriers in the test dataset, none were classified as intermittent carriers, 16/25 (64%) were misclassified as non-carriers and 9/25 (36%) as persistent carriers, adding further evidence that intermittent carriers are not distinct group, and a greater proportion are similar to non-carriers compared to persistent carriers.

We determined variable importance (i.e. how much each variable contributes to the prediction) by evaluating the mean decrease in accuracy (a measure of decrease in the model accuracy computed by permuting out-of-box error data) and the mean decrease in gini index (a measure of variance and resulting misclassification across the random forest nodes) after removal of each feature, i.e. taxon[38]. The top three features of importance by assessing the mean decrease in

accuracy were *S. aureus*, *Corynebacterium sp.*, and *S. epidermidis* (Fig. 5c). The top three features of importance by assessing the mean decrease in gini index were *S. aureus*, *Corynebacterium sp.*, and *D. pigrum*, with *S. aureus* clearly contributing the most to the model (Fig. 5d).

### *Staphylococcus aureus* phylogenetic associations with carriage

We next investigated if certain *S. aureus* lineages have a propensity for persistent nasal carriage or are more capable of dominance of the community compared to other competing resident bacteria. We used *S. aureus* isolate whole genome sequences with matched microbiome data (*n* = 172) and compared the *S. aureus* phylogenetic tree, and major multi-locus sequence types (MLST), to the colonisation state and the sample microbiome (Fig. 6a). Two clusters are defined at the bifurcation at the root of the phylogeny (Cluster A and B, Fig. 6a), as seen in large collections of diverse *S. aureus*[39]. There is a greater number of samples showing higher *S. aureus* abundance amongst isolates in cluster B (dominated by ST30, ST34, ST398, and ST45) with a lower number of samples showing higher abundance of species identified earlier as showing a negative association with *S. aureus* (Fig. 3a) than in Cluster A (dominated by ST5, ST8, ST15, ST7, and others). Matched CST data was available for 125 samples; 38/74 (51.4%) of cluster A compared to 33/51 (64.7%) cluster B samples were found in the *S. aureus* dominant CST I (Fig. 2). We examined differences in rarefied (i.e. per-sample normalised read data) *S. aureus* abundance (*n* = 111), which demonstrated a significantly higher abundance in samples in cluster B compared to cluster A (Mann-Whitney, *p* = 0.04) (Fig. 6b). Next, we assessed differences in Beta diversity between cluster A and B (Fig. 6c, d), and found a small but statistically significant (PERMANOVA analysis ($F(2) = 2.33$, $p = 0.04$) divergence of these groups. This suggests that *S. aureus* abundance and the associated microbiome (when *S. aureus* is present) is to some degree lineage specific.

## Discussion

Despite the importance of *S. aureus* colonisation as a risk factor for *S. aureus* infection, there is still only a limited understanding of what determines nasal *S. aureus* colonisation. In this work, we combine for the first time, large-scale microbiome sequencing with longitudinal culture data that, since the 1940's[32], has been used to define *S. aureus* colonisation. We have generated multiple new insights into the nasal microbial community structure of the anterior nares, substantially extending previous smaller-scale studies[3,17,31,40,41]. Like the previous study of older twins from Denmark[17], we identify seven community state types (CSTs), but with a different species composition which suggests that either the previous smaller study was unrepresentative or there is variation in the nasal microbiota even between two northern European countries. Importantly, our analysis of the seven CSTs revealed that women are more likely to have either CST VI (*C. accolens*) or CST VII (diverse group), suggesting an influence of sex on the wider nasal microbiome composition, as with *S. aureus* persistent colonisation[12] and load[17]. These novel insights provide important insights for better understanding microbial interactions, *S. aureus* colonisation resistance, and biotherapeutic targeting.

We demonstrate that there is a clear distinction in the microbial community structure that underlies persistent *S. aureus* carriage compared to non-*S. aureus* carriage in a large sample of individuals (Fig. 6e). We found that persistent carriage of *S. aureus* is negatively associated with three *Corynebacterium* species (including *C. jeikeium*, *C. accolens* and an unnamed *Corynebacterium sp*), *D. pigrum*, *S. epidermidis*, and *M. catarrhalis*. Notably, *C. jeikeium*, *C. accolens*, *M. catarrhalis*, and the unnamed *Corynebacterium sp*. have not been previously identified as negatively associated with *S. aureus* abundance in microbiome data[3,17,31]. We also find that the diverse CST VII partitions into smaller sub-clusters, found in multiple individuals and dominated by single species (Supplementary Fig. S8); these have most likely

become apparent given the scale of this study. We failed to replicate the negative association with *S. aureus* abundance with *Simonsiella sp.* or *Cutibacterium* (formerly *Propionibacterium) granulosum* as previously reported[17]. Neither species were found in our pre or post-QC data, suggesting that these are either uncommon species in England, perhaps only present in certain environmental conditions, or contaminants[29]. We also did not identify an association of *Finegoldia magna* and *Staphylococcus lugdunensis*, with *S. aureus*[42,43]; both organisms were identified within the dataset but filtered out due to strict contaminant removal criteria (species with an abundance <0.1% across all samples) which reflects a threshold determined from previous work, below which species were not reliably identifiable or distinguishable from contaminants[44]. Although these may be able to act as biologically significant determinants of carriage in a small number of individuals, we do not identify them at scale, which again may reflect true population level differences between studies or that these are contaminants.

In contrast, our findings did replicate the previously reported negative association of *S. aureus* with *Dolosigranulum* spp[17]. and *D. pigrum*[16]. Interactions between *D. pigrum and Corynebacterium* spp. and particularly *C. accolens* have recently been explored in vitro; *D pigrum* was shown to inhibit a single strain of *S. aureus* directly, whilst *C. pseudodipthericum*, *C. accolens*, *C. propinquum* conditioned media enhanced the growth of *D. pigrum*, and that this growth enhancement was not reciprocal[16]. *C. accolens* was found to both enhance the growth of *D. pigrum* through an unknown mechanism(s), and inhibit growth by processing host tri-acylglycerols into fatty acids with antibacterial properties[16]. Our network analysis further highlights the extent of the co-occurrence between taxa that are negatively associated with *S. aureus* carriage such as *D. pigrum* and multiple *Corynebacterium* spp. (Fig. 3b). Our data highlights further consistent co-occurrences, such as *S. aureus* with *M. osloensis* and *D. pigrum* with *Moraxella catarrhalis*, in both network groups and heatmap clustering, underscoring the robustness of these specific associations. We identify key central species within distinct subcommunities which may influence community assembly through metabolic interactions or niche modulation. Experimental validation via targeted co-culture studies will be essential to confirm these predicted interdependencies and clarify their mechanistic basis.

We found both *C. jeikeium* and *C. accolens* are negatively associated with *S. aureus*. While *C. jeikeium* has not been previously reported to inhibit *S. aureus*, *C. accolens* has been demonstrated to inhibit *S. aureus* experimentally[45,46], and as noted above, *C. accolens* can support or inhibit growth of *D. pigrum* - which can inhibit *S. aureus* directly[16]. This suggests *C. accolens* can inhibit *S. aureus* through direct or indirect means. This is supported by our data, whereby most individuals found in CST VI are dominated by *C. accolens* and have a low abundance of *S. aureus* (Fig. 2a). Notably, women have an increased relative likelihood for CST VI; this provides a mechanistic explanation for why women are less likely to be persistent carriers of *S. aureus*, although this is unlikely to explain the entire variation in colonisation rates between men and women[12]. As discussed previously, *C. accolens* has also been reported to be positively associated with *S. aureus* in a smaller microbiome study and was shown to promote the growth of *S. aureus* in vitro[3]. In our data, *C. accolens* is found at a lower abundance in multiple CSTs, including co-occurrence with *S. aureus* in a small sub-cluster within the *S. aureus* dominated CST I (see CST I in Fig. 2a). This suggests that there is a strain level or lineage variation in *C. accolens* and/or *S. aureus* that contributes to this discordant relationship – which again highlights the need for large scale studies. An alternate possible explanation would be of other colonising species shaping in vivo interactions between *S. aureus* and *C. accolens* (high-order interactions); studying associations between *S. aureus* and group of species (instead of one) would help resolve this hypothesis.

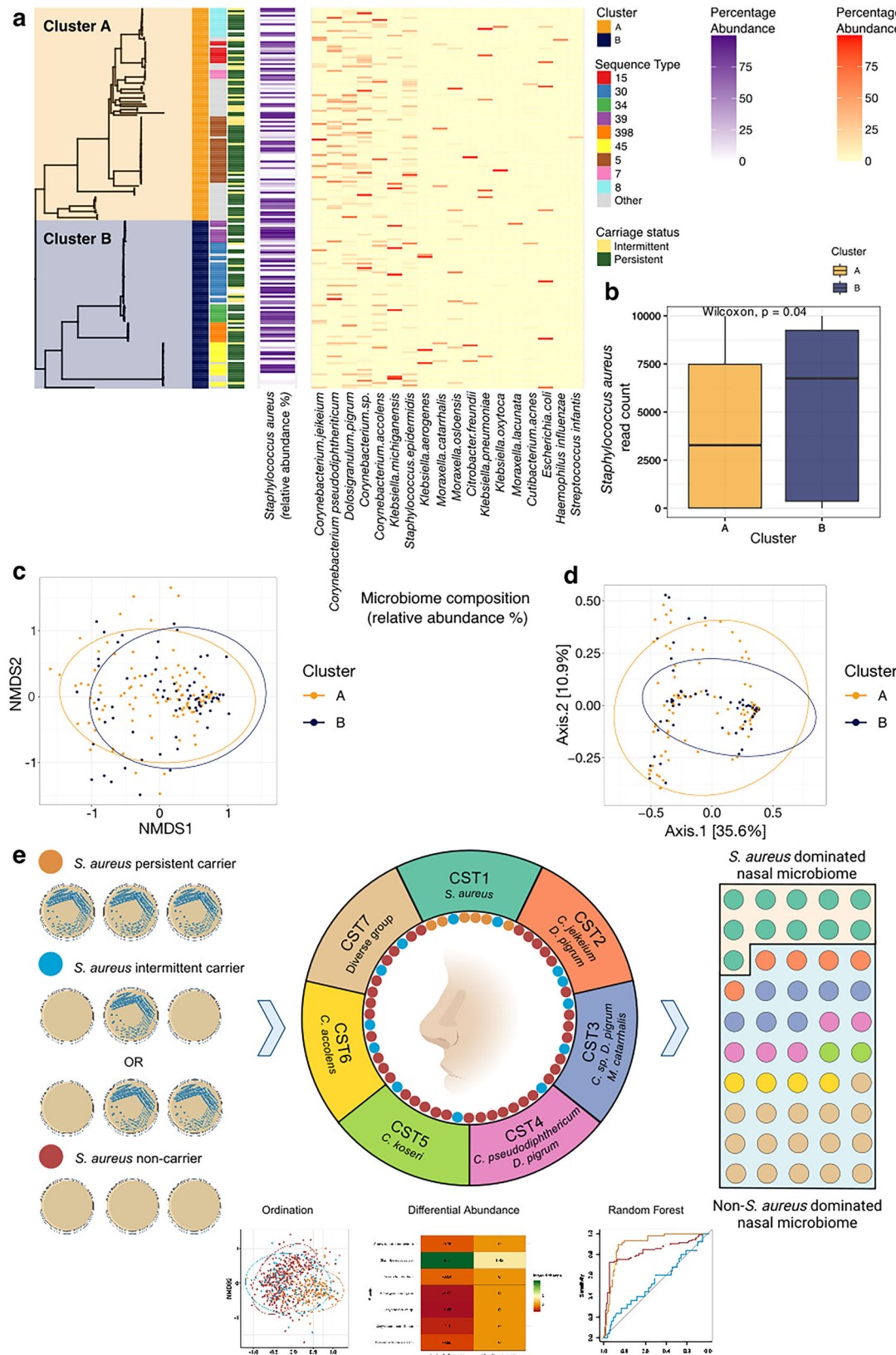

Our data identified a negative association between *S. epidermidis* and *S. aureus*, corroborated by the differential abundance analysis and the random forest model. Although negative[31] and positive[17] associations have been reported previously, similar to *C. accolens*, a previous study has shown that a certain proportion of *S. epidermidis* strains secrete a serine protease, Esp, which inhibits *S. aureus* growth and biofilm production[19]. Human nasal *S. epidermidis* isolates have been shown to produce peptides with antimicrobial activity against nasal *S. aureus* isolates[47]. Furthermore, the effect observed could also be explained by environmental factors. *S. aureus* requires a higher relative humidity (87%) than *S. epidermidis* (81%), and *S. epidermidis* may be more commonly found in drier noses[48]. Our finding that *M. catarrhalis*

**Fig. 6 | Variation of the anterior nares microbiome with the *Staphylococcus aureus* phylogeny. a** Maximum-likelihood tree of *S. aureus* whole-genome sequences cultured from persistent and intermittent carriers labelled with their associated carriage status, sequence-type and microbiome. **b** Box plots comparing rarified *S. aureus* abundance by cluster: cluster A, $n = 62$; cluster B, $n = 49$. Each data point is derived from a nasal sample from a distinct individual. The midline of the boxplot represents the median value; the lower limit of the box represents the first quartile (25th percentile), and the upper limit of the box represents the third quartile (75th percentile); the whiskers (upper and lower) extend to the largest and smallest value from the box, no further than 1.5*IQR from the box. Statistical significance from pairwise comparisons tested using the Wilcoxon rank-sum test (two-

sided), ($p = 0.04$). **c**, **d** Ordination plots representing Beta diversity by Bray-Curtis distance and coloured by the phylogenetic clusters (A or B) representing the bifurcation of the tree. Bray-Curtis distance between samples representing phylogenetic clusters A and B samples significantly differed, although weakly, by PERMANOVA analysis ($F(2) = 2.33$, $p = 0.04$). Beta dispersion (PERMDISP) analysis showed no significant difference in dispersion between groups (F = 1.26, $p = 0.29$). **c** shows an NMDS plot **d** shows a PCoA plot. Data ellipses represent the 95% confidence level that values lie within this space assuming a multivariate t-distribution. **e** Graphical representation of abstract created in BioRender. Ng, D. (2025) https://BioRender.com/o73y544.

is negatively associated with *S. aureus* in adults is new, though abundance has been reported to be inversely correlated with *S. aureus* in children[18].

As would be expected, persistent carriage was positively associated with *S. aureus*. Strikingly, we found that in ~50% of persistently colonised individuals *S. aureus* is the single most abundant organism in the nasal microbiome, representing >75% of reads for ~35% persistent carriers. This domination is reflected in reduced Alpha diversity among persistent carriers compared to non-carriers and is further supported by the greater stability of Alpha diversity amongst the persistent *S. aureus* carriers compared to non-carriers. This suggests that *S. aureus* may act as a keystone species which principally determines its own continued carriage and can suppress other members of the nasal microbiome. While most nasal microbiome research has concentrated on how other members of the nasal microbiome prevent *S. aureus* colonisation, our data indicates that future investigations should now focus how *S. aureus* excludes other species from the nasal microbiome and is resistant to antagonistic compounds or growth conditions generated by competing species such as *Corynebacterium* spp. and *D. pigrum*.

The subset of very high *S. aureus* load carriers we identified might also be of clinical importance. The high bacterial load of *S. aureus* and a lack of antagonistic species amongst persistent carriers specifically may explain why these individuals are more likely to become infected by *S. aureus*; inoculation of a wound with a high load of *S. aureus* in the absence antagonistic species (that might inhibit *S. aureus*) could make infection is more likely. Future studies which make the use of nasal microbiome-based stratified participants, and/or measure *S. aureus* as a quantifiable trait (e.g. quantitative PCR), are required to understand the consequences of high-load *S. aureus* dominant nasal colonisation on the rest of the human microbiome (e.g. skin and gut[26]), subsequent risk of infection, and transmission.

Importantly, our study demonstrates that *S. aureus* carriage can be predicted from microbiome data with a moderate degree of accuracy. Notably, the model is more sensitive in predicting *S. aureus* non-carriage. This is particularly important given a single swab of the anterior nares is limited in its diagnostic accuracy[34]. Further, the high negative predictive value of the model for persistent carriage we present here, using the microbiome data from a single swab, may improve the identification of true negatives in *S. aureus* screening and identify those who are unlikely to be persistently colonised, facilitating a selective approach to patient decolonisation. Further large-scale studies using higher resolution metagenomics and clinical data, will likely significantly improve this, whilst comparison with *S aureus* qPCR will be useful for moving towards being able to define risk of *S. aureus* infection based on a single swab.

We have determined that intermittent carriers do not have a distinct microbial community. The lack of biological relevance pertaining to culture-defined intermittent carriage was first proposed by Van Belkum et al. [10,49] who theorised that: (1) non-carriage is either incidental and most people are actually intermittent carriers or (2) intermittent carriers are non-carriers who have picked up *S. aureus* from the environment. Our data provides evidence that the latter of the two is

partly correct and provides new insights into this. Intermittent carriers, as defined by one or two culture positive swabs, belong to one of two populations: a population with a *S. aureus* dominated microbial community structure (similar to persistent carriers) with the absence of species identified as negatively associated with *S. aureus* or a population with one of several microbial community structures which are not dominated by *S. aureus*, are more diverse, and often dominated by other species (similar to non-carriers) (Fig. 6e). Therefore, the most parsimonious explanation is that intermittent carriers, given their low *S. aureus* abundance and predominance amongst non-carriage CSTs, are effectively *S. aureus* 'non-dominant carriers' who are only transiently colonised with *S. aureus*, for example due to environmental/household exposure/other body sites e.g. gut (i.e. hypothesis 2 proposed by Van Belkum *et al.*). While two swab positive intermittent carriers are '*S. aureus* dominant carriers' (akin to persistent carriers), reflected by their higher *S. aureus* abundance and greater representation amongst the *S. aureus* dominant CST, and therefore likely to be persistently colonised individuals that were negative by culture in one swab.

We identified a relationship between certain *S. aureus* lineages and *S. aureus* abundance and the associated microbiome. This suggests that *S. aureus* abundance and carriage is to some degree lineage specific. It is noteworthy that these colonising *S. aureus* sequence types are found readily amongst colonising and invasive isolates[50]. Previous studies of human experimental colonisation with *S. aureus* identified that after decolonisation and artificial inoculation, persistent carriers had higher loads of *S. aureus* than intermittent or non-carriers and were more likely to select their own strain[10]. Our data suggest that this might be due to lineage-specific effects of persistent carriers' strains being better adapted to persistent colonisation, enabling them to reach higher abundances. The same study also showed that persistent carriers had higher serum IgG and IgA levels to certain *S. aureus* antigens (SasG, TSST-1, SEA, ClfA, CHIPS). Given the known variation of mobile genetic element (MGE) content in *S. aureus* lineages (TSST-1, SEA, CHIPS are all MGE acquired), a propensity for lineages with particular MGE content to be found at higher abundance in persistent carriers may explain this variation in antibody levels. Overall, these lineage relationships are likely to represent adaptations of *S. aureus*[51] that impact host adaption to colonisation, transmission success and intra- and interspecies competition. This requires further investigation in a larger cohort of participants and to elucidate the mechanisms for this lineage specific adaption to colonisation. A shotgun metagenomic led approach would help understand the potentially confounding impact of mobile genetic elements on the relationship of *S. aureus* lineages and the resident microbes.

This study has potential limitations, including those inherent to 16S rRNA gene studies such as choice of referencing database; though we have previously demonstrated the accuracy of our sequencing and analysis pipeline[44], further validation of taxa (for example: the unnamed *Corynebacterium sp.* negatively associated with *S. aureus*) identified at a species level with selective culture and/or shotgun metagenomic sequencing will be useful. Further, although the study participants are healthy and sampled at home, the blood donor

cohorts used will still not be entirely representative of the population of England, which is itself clearly not representative of all global populations. Studies in different populations using standardised methods are required to explore this variation. Additionally, a minor variation in the calculation of CSTs may contribute, in part, to the differences seen in results with Liu et al. [17]. Next, recent work has highlighted the potential significance of gut colonisation of *S. aureus* which has been hypothesised to contribute to re-colonisation of the anterior nares, and pose relevance to clinical infection and transmission[27,52,53]. We do not examine gut colonisation and future work should aim to systematically examine within-host cross-niche transmission of *S. aureus* and re-examine the persistence of colonisation across multiple host niches in large and generalisable cohorts. Finally, a small proportion of culture-negative samples had a high *S. aureus* abundance. Several factors may explain this: (a) unfavourable transport/storage conditions for a small number of samples, (b) non-culturability of the *S. aureus* strain due to auxotrophism or being in a viable but nonculturable state, or (c) the presence of dead bacteria at the time of sampling due to exposure to antimicrobials or the action of the immune system.

In summary, we present the most comprehensive assessment of the microbial composition of the anterior nares to date. Our data provides multiple new insights and identifies key microbial interactions and variation that underpin the composition of the human nasal microbiome, and in particular colonisation by *S. aureus*.

## Methods

### S. aureus culture

After 10 seconds of vortexing, nasal swabs in Amies transport media (Medical Wire) were transferred to a tube containing 2 ml Tryptic Soy broth supplemented with 6.5% NaCl (Medical Wire) and incubated overnight at 37 °C, in air. The remaining Amies solution was transferred to an Eppendorf with 500 μl of glycerol, pipette mixed and stored at -70 °C. 10 μl of the overnight enrichment broth was streaked onto chromogenic Staph Brilliance 24 agar plates (Oxoid) and incubated overnight at 37 °C. If no blue colonies were identified after 24 hours of incubation, the plate was returned to the incubator overnight and rechecked. Blue colonies are with the phenotypes of putative *S. aureus* were sub-cultured onto Columbia Blood agar (5% horse blood) and incubated overnight at 37 °C. Colonies from these plates were inspected visually for phenotype indicators, and tested for coagulase and protein A via latex agglutination test (Pro-Lab Diagnostic). Where there were queries or discrepancies, species level identity was confirmed using Matrix assisted laser desorption and ionisation – Time of Flight (MALDI-ToF). All isolates were stored (Pro-Lab Diagnostics) at -80 °C.

### DNA processing and 16S rRNA gene polymerase chain reaction

Prior to extraction, residual sample transport medium from nasal samples was stored at -70 °C in -33% v/v glycerol. Total DNA was extracted from nasal sample transport medium after an additional mechanical lysis step (Lysing matrix E, MP Biomedicals) either via the MPBio MPure-12 instrument, (MPure Bacterial DNA Kit, MP Biomedicals) or manually using the FastSpin Kit for Soil (MPBiomedicals), including the heated elution step. DNA was then stored at -70 °C until use. V1V2 specific primers with attached sequencing adaptors and indexes (Table S1) were used for PCR to amplify the bacterial 16S ribosomal gene regions[54]. All primers were purchased from Sigma-Aldrich. V1V2 region was selected as it preferentially differentiates between important nasal microbiota such as *S. aureus* and coagulase-negative staphylococci, and more accurately determines upper and lower respiratory microbiomes, when compared to other variable regions of the 16S rRNA[55–57]. PCR amplification mastermixes were prepared manually using a Q5 High-Fidelity Polymerase Kit (M0491, New England Biolabs). PCR amplifications were setup in triplicate (25ul each), products were pooled into a single volume per sample, and all samples were subsequently purified using an AMPure XP (Beckman Coulter) workflow at a ratio of 0.8X. Libraries were quantified using the Qubit HS DNA Kit (ThermoFisher). Equimolar pools were then created. Negative controls included a sample extraction control, a PCR water control, and an aliquot of the glycerol used for storage, whilst a positive control was represented by purified water spiked with *S. aureus* DNA.

### DNA sequencing

Per experiment, an equimolar pool of PCR libraries was sequenced at the Wellcome Sanger Institute in-house sequencing facility, using the Illumina MiSeq (300 bp paired-end reads, v3 Reagent Kit). Accession numbers for the sequencing data is in Supplementary Data 1.

### 16S rRNA gene sequence quality control and taxonomy assignment

We used a modified mothur MiSeq standard operating procedure (SOP) to process paired fastq files (MOTHUR wiki at http://www.mothur.org/wiki/MiSeq_SOP)[58]. The four poly(NNNN)s present in the adapter/primer sequences of contigs assembled with the make.contigs command in mothur were trimmed with the PRINSEQ program, before the modified MiSeq SOP was resumed. The Silva bacterial database 'silva.nr_v132.align' was used to align quality-screened sequences and chimeras removed using Uchime[59]. Sequences were then classified using the same Silva reference database and the Silva taxonomy database silva.nr_v132.tax, with the removal of chloroplast, mitochondria, unknown, and eukaryota sequences. We clustered high-quality unique sequences with Oligotyping v2.1[60] (-M option to 1000), which were assigned to NODES, and referred to as operational taxonomic units (OTU) from here, with the Minimum Entropy Decomposition (MED) option (Supplementary Data 3). We created a customised silva SSU Ref database (NR99, release 132), where we removed the majority of environmental and uncultured taxa, and carried out taxonomic assignment with ARB (v6.0.6-3)[61] (Supplementary Data 4). In some instances, where a mismatch was observed within the taxonomic groups, we assigned taxa to the OTU sequence with BLAST[62] (see Supplementary Table S2). We then combined the output in R (v4.4.1) into a phyloseq[63] object for onward analysis.

### Contaminant removal and accounting for variability in sequencing depth

We identified contaminants and removed these by identifying batch effects and accounting for negative controls[28–30]. Batch effects were assessed by calculating the spearman's correlation co-efficient of species against each location of extraction, and location of PCR reaction. We then examined correlation of species with sample DNA concentrations. We used well characterised kitome and environment contaminants to identify additional associated contaminants by calculating species-species correlation coefficients. We used Decontam v1.16.0[64] to account for laboratory negative controls, run with the isnotcontam function and with each sequencing run provided as a batch (further details below and in Supplementary Fig. S3 and S4 and Supplementary Table S3).

We determined a suitable rarefication depth of 10,000 reads using rarefaction curves and examining the read depth at which the majority of sample taxa numbers plateaued (see rarefaction section below and Supplementary Fig. S5). We removed species with an abundance of less than 0.1% across samples, below which we expected the removal of most contaminants and account for the variability in rare species composition between runs[44]. For diversity analyses, the rarified dataset was used. For abundance analyses, to mitigate data loss, we combined samples with greater than 500 high quality reads with samples that had greater than 10,000 reads and rarefied. (see rarefaction section, Supplementary Fig. S3, S4, S5 and Supplementary Table S3).

## Identification of contaminants

Removal of taxa below the 0.1% threshold resulted in 115/2,322 OTUs remaining. The samples were processed over two time periods. Over the first time period ($n = 1,099$), there were five locations for DNA extractions and five locations for PCR amplification. We used spearman's correlation coefficient to identify batch effects, specifically species with abundance that was associated with the extraction and PCR locations (Supplementary Table S3, Supplementary Figs. S3 and S4). For the second time period ($n = 767$), extractions and PCR amplifications took place in one location and therefore batch effects by location was not examined. We used spearman's correlation coefficient to identify taxa that correlated with PCR qubit values (post-PCR amplification DNA concentration); previously, lower sample DNA concentrations have been associated with contaminants[30,65] (Table S3, Supplementary Fig. S3 and S4). We used hierarchical clustering to identify species that clustered with one another, which allowed for the identification of taxa that were correlated with well-characterised and suspected contaminants[30] (Supplementary Table S3, Supplementary Fig. S3 and S4). As a final check, we used the R package Decontam (v1.16.0)[64] to account for negative controls, with each sequencing run considered as a batch (Supplementary Table S3, Fig. S3 and S4).

## Determining a rarefication threshold

We subset CARRIAGE samples and generated rarefaction curves for samples with greater than 1000, 5000, 10,000, 15,000, 20,000, 100,000 high-quality reads respectively (Supplementary Fig. S5). In order to determine a rarefaction threshold, we identified the slope of each rarefaction curve at the respective high-quality read threshold using the rareslope() function in phyloseq (Supplementary Fig. S5); given the large dataset, visualising the point at which the curves plateaued was not possible. It was apparent that at greater high-quality read thresholds, a larger proportion of the samples reached a (near) plateau. We aimed to reach a balance between losing a large number of samples and retaining a dataset where the rarefaction curves for the vast majority of samples had plateaued; this was met at 10,000 reads (Supplementary Fig. S5). From here, we either use the dataset rarefied to an even-depth to the minimum read count above this threshold (10,004) or this dataset combined with samples with greater than 500 reads but less than 10,000 reads, to minimise data loss and consistent with previous analyses[54].

## Diversity analysis

We conducted microbial diversity and compositional analysis in R using diversity indices calculated with the phyloseq (v1.40)[63] and vegan (v2.6-4)[66] packages. Alpha-diversity indices (Shannon's and Simpson's) were calculated on rarefied read counts. Sample microbial composition is consistently represented with relative abundances. We used Principal Coordinate Analysis (PCoA) and Non-Metric Dimensional Scaling (NMDS) with the bray–curtis distance matrices to visualise differences in sample diversity by condition (e.g. *S. aureus* colonisation status).

## Data visualisation and statistical analysis

We manipulated data in Excel 2016 and R (v4.4.1). We generated figures using ggplot2 (v3.4.0), phyloseq (v1.40)[63], ComplexHeatmap (v2.24.1)[67], microViz (v0.11.0), and ggtree (v3.16.3)[68]. We evaluated differences in Alpha indices with Mann-Whitney-U and Kruskall-Wallis tests where appropriate. We used PERMANOVA to estimate differences between Bray-Curtis distances observed by study groups with the vegan package (v2.6-4)[66]. Beta dispersion was assessed using PERMDISP based on distances from group centroids, followed by Tukey's HSD for pairwise comparisons. To determine the number of clusters in the data, we calculated a gap statistic with ordination values using Bray-Curtis distances, using the R package cluster function clusGap() (Supplementary Fig. S7). We investigated the association of plausible lifestyle and comorbidities risk factors with Community State Types (CST) using a multinomial logistic regression model analysis (CST ~ sex + smoking status + pet ownership + healthcare contact + chronic skin condition + asthma + allergies). We used ANCOM-BC (v1.6.4)[69] to evaluate differential abundance of microbial species in the study groups; we used the ancombc2 function with default settings, but specified taxa with a prevalence of less than 0.1% to be removed and a library cut-off of 500 reads, and provided a non-rarefied count table as a centred log ratio transformation is conducted[69]. Species-level networks were inferred with NetCoMi (v1.2)[70] using SparCC correlations (zeroes replaced by a pseudocount; centred log-ratio transform; 1000 bootstraps), then analysed for centrality and community structure via fast-greedy clustering. The network was visualised with igraph (v2.1.4). This study complies with the STORMS guidelines[71] for reporting.

## Random forest model

We trained two separate models, one utilising all samples above 500 reads where samples with greater than 10,004 had been rarefied ($n = 1055$), and another including the rarefied dataset alone ($n = 795$). The rarefied dataset performed better compared to the combined dataset (see Supplementary Results for further details). We used the R package randomForest (v4.7-1.1)[38] to fit a random forest classifier for carriage status (relative_microbial_abundance_data ~ carriage_status). The model was trained using a randomly subsampled dataset of the microbial features (in relative abundance format) representing 80% of the data (ntrees=1000), and tested on the remaining 20% to evaluate model robustness. We chose the number of predictors sampled for splitting at each node (mtry) with the tuneRF() function. We obtained sensitivity and specificity values of the model with the R package caret (v6.0-90) whilst receiver operating characteristic curve (ROC) curves and AUC were obtained with the R package pROC (v1.18.4). *P*-values less than 0.05 were considered statistically significant.

## Whole Genome Sequencing of *Staphylococcus aureus* isolates

*S. aureus* isolates were sequenced at the Wellcome Sanger Institute with 96 sample libraries sequenced on a 300 bp PE MiSeq lane (with a 1% PhiX spike). European Nucleotide Accession number for isolates is presented in Supplementary Data 2.

## Phylogenetic analysis

From the raw whole genome sequencing data, we generated quality control metrics, and trimmed reads, with the nextflow pipelines, bacQC (github.com/avantonder/bacQC). Species classification for each sample was performed using Kraken and Bracken[72]. We discarded samples with less than 90% reads matching to *S. aureus* and those with <30x coverage from onward analyses. Using the nextflow pipeline, assembleBAC (github.com/avantonder/assembleBAC), we produced annotated assemblies with trimmed fastqs. The pipeline uses shovill (v1.1.0) for assembly. We annotated assemblies with prokka (v.1.14.5)[73] using a genus-specific database from RefSeq for annotation. Assemblies with an N50 value < 10,000, length of less than 2.6Mbp or greater than 3.0Mbp, or with a spuriously high number of contigs summarised by MultiQC[74] and the QC metrics generated by Panaroo(v1.3.4)[75] were removed from onward analyses. Samples with greater than 300 contigs were found to be outliers.

We assigned sequence types (STs) with mlst (v2.19.0) (github.com/tseemann/mlst), and where these were not assigned, assemblies we queried the sequences on the PubMLST web server[76]. We produced core-genome alignments with Panaroo (v 1.3.4)[75] with a core-genome threshold set to 98%. We extracted variant sites from the core-genome alignment with snp-sites (v2.5.1)[77] and coupled with associated values for invariant sites to build a maximum likelihood (ML) phylogenetic tree. We used IQ-TREE (v2.1.2)[78] to estimate ML phylogenetic trees with the optimal phylogenetic trees determined by ModelFinder[79] and

branch support statistics generated using the ultrafast bootstrap method[80].

## Statistics & Reproducibility

No statistical method was used to predetermine sample size. Randomisation or blinding was not appropriate for these experiments. Data excluded from analyses included duplicate samples from individuals where appropriate, contaminant sequences, and through rarefaction of high-quality reads or a predetermined high-quality read count threshold (500 reads) to account for sample sequencing quality and normalisation, as described above.

## Ethics approval and consent to participate

The CARRIAGE study protocol was approved by the National Research Ethics Service Committee North-West - Lancaster Research Ethics Committee, 27/06/2016, REC reference: 16/NW/0507, IRAS project ID: 202688. All participants provided informed consent. The study is registered at ISRCTN: ISRCTN10474633.

## Reporting summary

Further information on research design is available in the Nature Portfolio Reporting Summary linked to this article.

## Data availability

The sequencing data generated in this study have been deposited in the European Nucleotide Archive (ENA) under the accession codes listed in Supplementary Data 1 and 2 (https://www.ebi.ac.uk/ena). The raw sequencing reads and associated metadata are publicly available without restriction. The processed taxonomic tables used for analysis are provided as Supplementary Data 3 and 4. Unique sequences ('Nodes') generated from the bioinformatics pipeline have been deposited in Zenodo (https://doi.org/10.5281/zenodo.17160106). The personal data for the CARRIAGE study participants is not publicly available due to restrictions on data release. The participant consent form states that study data will be accessible to researchers who have relevant scientific and ethics approvals for their planned research. As per the study protocol applications can be made to access the data which are reviewed by the Data Access Committee at the Cardiovascular Epidemiology Unit, Cambridge University. Informal enquiries can be made to Dr Ewan Harrison (eh439@cam.ac.uk). To request the application form please email CEU-DataAccess@medschl.cam.ac.uk. The full study protocol outlines this: https://wellcomeopenresearch.org/articles/10-405 [81].

## Code availability

The authors declare that all data cleaning, analysis and visualisation associated with this article were performed using previously published methods, the applications of which are appropriately cited in the corresponding sections in the Methods. No custom code was developed for the aforementioned purposes.

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

## Acknowledgements

We wholeheartedly thank the CARRIAGE study participants for taking part in the CARRIAGE study and providing the samples that were critical to being able to conduct this research. This work was supported by Wellcome Collaborative Award in Science (Grant no. 211864/Z/18/Z) to S.J.P., J.P., J.D., J.A.G., E.M.H., Isaac Newton trust Grant 17.07(1) to EMH, UKRI Innovation Fellowship: MR/S00291X/1 to EMH, Wellcome Grant reference: 220540/Z/20/A, 'Wellcome Sanger Institute Quinquennial Review 2021-2026' – core funding of Wellcome Sanger Institute, Wellcome Clinical PhD Fellowship: 222903/Z/21/Z to DAg. This research was supported by the NIHR Cambridge Biomedical Research Centre (NIHR203312*). The epidemiological coordinating centre of the CARRIAGE study was additionally supported by awards from the NIHR Blood and Transplant Research (5Unit (BTRU) in Donor Health and Behaviour (NIHR203337), NIHR Cambridge BRC (NIHR203312) (*), and by Health Data Research UK (HDRUK2023.0028), which is funded by the UK Medical Research Council, Engineering and Physical Sciences Research Council, Economic and Social Research Council, Department of Health and Social Care (England), Chief Scientist Office of the Scottish Government Health and Social Care Directorates, Health and Social Care Research and Development Division (Welsh Government), Public Health Agency (Northern Ireland), British Heart Foundation and Wellcome. DAg, Clinical Lecturer, CL-2024-21-002, is funded by NHS England/NIHR for this research project (*). JD holds a British Heart Foundation Personal Chair (CH/12/2/29428). *The views expressed are those of the authors and not necessarily those of the NIHR, NHS, or the Department of Health and Social Care'.

## Author contributions

Conceptualisation: S.J.P., E.M.H., J.P., J.D., J.A.G., and D.Ag. Methodology: D.Ag., M.C.d.G., K.L.B., J.W., S.J.S., J.P., S.J.P., E.M.H. Data curation: D.Ag., K.L.B., B.B., S.K., S.B., R.H., C.P., M.R.W., C.M., S.T.G., C.R.d.S., L.S., J.B., S.D., E.J., M.J., D.An., S.I., A.M. Investigation: K.L.B., B.B., K.E.R., P.N., S.T.G., C.R.d.S., L.S., A.A., L.L., C.R., X.B., J.B., D.Ag. Software: D.Ag., M.C.d.G., A.J.v.T. Resources: J.D., A.S.B., E.D.A., M.H., S.J.P., E.M.H. Formal analysis: DAg. Validation: DAg, MCdG. Visualization: DAg, DYKN (Biorender images). Writing—original draft preparation: D.Ag., E.M.H. Writing—review and editing: All authors. Project administration: K.L.B., B.B., D.Ag., S.B., R.H., C.P., M.R.W., C.M., C.C., S.D., E.J., M.J., D.An., S.I., A.M., S.J.P., E.M.H. Supervision: E.M.H., S.J.P., J.P., J.D., M.C.d.G., J.W. Funding acquisition: E.M.H., S.J.P., J.P., J.D., J.A.G., D.Ag. Where author initials are the same: DAg, Dinesh Aggarwal; DAn, David Anderson.

## Competing interests

A.S.B. reports institutional grants outside of this work from AstraZeneca, Bayer, Biogen, BioMarin, Bioverativ, Novartis, Regeneron and Sanofi. J.D. serves on scientific advisory boards for AstraZeneca, Novartis, and UK Biobank, and has received multiple grants from academic, charitable and industry sources outside of the submitted work. The remaining authors declare no competing interests.

## Additional information

**Dinesh Aggarwal** [1,2,3] ✉, **Katherine L. Bellis** [1,2], **Beth Blane**[1,2], **Marcus C. de Goffau**[2], **Josef Wagner**[2], **Duncan Y. K. Ng** [2], **Kathy E. Raven**[1], **Plamena Naydenova**[1,2], **Stephen Kaptoge**[4,5], **Susan Burton**[4], **Rachel Henry**[4], **Catherine Perry**[4,5], **Matthew R. Walker** [4,5], **Carmel Moore**[4], **Carol Churcher**[1], **Sophia T. Girgis** [1,2], **Catarina Ribeiro de Sousa**[1,2],

Lauma Sarkane[1,2], Joe Brennan[1,2], Asha Akram[1,2], Shannon Duthie[4,6], Elisha Johnson[4,6], Mercedesz Juhasz[4], David Anderson[4], Susan Irvine[4], Amy McMahon[4,6], Liz Lay[7], Susannah J. Salter[7], Claire Raisen[7], Xiaoliang Ba[7], Mark Holmes[7], Andries J. van Tonder[7], Emanuele Di Angelantonio[4,5,6,8,9,10], Adam S. Butterworth[4,5,6,8,9], Joan A. Geoghegan[11,12], John Danesh[4,5,6,8,9,13], Julian Parkhill[7], Sharon J. Peacock[1] & Ewan M. Harrison[1,2] ✉

[1]Department of Medicine, University of Cambridge, Cambridge, UK. [2]Parasites and Microbes Programme, Wellcome Sanger Institute, Hinxton, Cambridge, UK. [3]Department of Infectious Diseases, Imperial College London, London, UK. [4]British Heart Foundation Cardiovascular Epidemiology Unit, Department of Public Health and Primary Care, University of Cambridge, Cambridge, UK. [5]Victor Phillip Dahdaleh Heart and Lung Research Institute, University of Cambridge, Cambridge, UK. [6]National Institute for Health and Care Research Blood and Transplant Research Unit in Donor Behaviour, University of Cambridge, Cambridge, UK. [7]Department of Veterinary Medicine, University of Cambridge, Cambridge, UK. [8]British Heart Foundation Centre of Research Excellence, University of Cambridge, Cambridge, UK. [9]Health Data Research UK Cambridge, Wellcome Genome Campus and University of Cambridge, Cambridge, UK. [10]Health Data Science Research Centre, Human Technopole, Milan, Italy. [11]Institute of Microbiology and Infection, University of Birmingham, Birmingham, UK. [12]Department of Microbes, Infection and Microbiomes, College of Medicine and Health, University of Birmingham, Birmingham, UK. [13]Human Genetics Programme, Wellcome Sanger Institute, Hinxton, Cambridge, UK. ✉e-mail: d.aggarwal@imperial.ac.uk; eh6@sanger.ac.uk

