## [Transparent Peer Review file · Nature Communications]

Large-scale characterisation of the nasal microbiome redefines *Staphylococcus aureus* colonisation status

Corresponding Author: Dr Ewan Harrison

Version 1:

Reviewer comments:

Reviewer #1

(Remarks to the Author)

This is an interesting large-scale study on the human nasal microbiome that was actually missing in the microbiome field. It largely extends our knowledge on nasal microbiome composition and it provides many new findings, e.g on microbiome types, and *S. aureus* association with other nasal bacteria and host sex.

I am not an expert in computational biology and statistics. Nevertheless, I got the impression that the provided data could yield more important findings if interpreted more extensively, thereby further increasing the novelty and impact of the study.

1. Nasal microbiomes are analyzed by 16S sequencing, which is often regarded as yielding information only on the genus but not on the species level. Nevertheless, the authors present abundances of species. They obviously amplified other 16S regions (V1-V2) than previous studies, they refer to (V1-V3 or V3-V6). The reasons for their choice and potential relevance for differences to findings in the other studies should be explained in detail in the Results section and they might be addressed in the discussion.

2. The authors refer to the 7 community state types previously defined by Liu et al but they obviously use quite a different definition for their 7 CSTs. The CST definition of Liu et al should be described in the introduction and the differences in the present study, in particular dominance by other bacterial species, should be mentioned in more detail. I also suggest using other names for the CSTs (e.g. A-G instead of 1-7) to avoid confusion. Several rare CSTs were also found and combined in the new CST7. It would be interesting to point out how rare these unusual CSTs are and how they are dominated.

3. 16S data are correlated with *S. aureus* culture data but how the abundances correlate is not addressed. What percentage of *S. aureus* culture-negative samples have *S. aureus* 16S rDNA (or the other way around)? Previous studies have suggested that the majority of human nasal microbiomes contain at least some *S. aureus* 16S but that culture positivity depends on a specific bacterial threshold density. Can such a threshold be defined? This would be highly important for the future use of 16S-based diagnostics.

4. The data could be used more extensively to describe positive or negative associations between nasal bacterial species that could instruct detailed functional studies in the future. Considering the size of the dataset, the analysis could go beyond pairs of species, towards combinations of species.

5. Previous studies have demonstrated antagonistic interactions of *S. aureus* with species such as *Finegoldia magna*, *Staphylococcus lugdunensis*, and others. Do the authors find similar associations? Even if they cannot confirm these data it would be valuable to address the point.

Specific points:

6. Reconsider the title. It is known that the nasal microbiome defines the capacity of *S. aureus* to colonize but how does it redefine colonization?

7. Abstract, lines 57-58: defining the *S. aureus* colonization statuses should require only the detection of *S. aureus* 16S DNA but machine learning could help to predict its stability (persistent vs. intermittent) right?

8. Line 59: "likely better adapted" than what?

9. Results, lines 74-79: the logic in this sentence is not fully clear. How do all these points support the notion that there are only two categories? Moreover, the sentence implies that intermittent carriers do not exist but the authors probably mean that intermittent carriers are not a distinct community state type.

10. Line 81: please specify "colonization", colonization rate or *S. aureus* nasal abundance?

11. Line 82: better "...though many of these studies are based on only a small number of study participants..."

12. Line 91: "lower overall diversity" compared to?

13. Line 97: add "sp" or "spp" to genus names

14. Line 98: "lower overall bacterial density" compared to? "16S rRNA gene"
15. Line 106: What is meant with "positive integration"?
16. Line 154: define QC
17. 159-160: Remove one „S. aureus“
18. Lines 165-157: sounds redundant with the sentence before
19. 170-171: Regarding the variance within their groups, better also add the PERMDISP statistical results, at least in the figure legends.
20. 172-173: wrong figure cited (1G)
21. Lines 179- 181: sounds weird – S. aureus culture positivity should be associated with both, persistent and intermittent carriage
22. Lines 186-187: sounds contradictory with the sentence before.
23. Lines 194-195: make clear that this statement refers to a new CST classification that differs considerably from the previous CST scheme. Explain how the new CSTs were defined (like which threshold for assuming different CSTs).
24. 203: "pseudodiphtheriticum"
25. Lines 207- 210: sounds again unclear and potentially contradictory to the statements in the paragraph before.
26. Line 230: abundance of what?
27. Line 232-233: "positive association of S. aureus with persistent carriage" sounds trivial.
28. Line 245: what is meant with "pairwise Alpha diversity"?
29. Line 251: "...the majority of microbiomes of intermittent carriers clustered with those of..."
30. 263: Wrong figure cited (3E)
31. 280: please rephrase, refer to point 9 above.
32. Three paragraphs starting at line 183: please address how the abundance of S. aureus 16S compares to culture positivity. To my understanding, overall microbiome composition should help to predict the stability of S. aureus colonization while the presence of S. aureus should be deduced from the detection of its rDNA (see my point 7)? But then the authors state in line 300 – 301 that their model was unable to predict intermittent colonization. Please explain better, I am confused...
33. Line 323-329: difficult to understand, please rephrase. How are the two clusters defined? Mention also the dominating STs in cluster A. The whole paragraph would profit from a clear conclusion. It would be helpful also to state how the identified nasal STs compare to the abundant invasive STs in the UK.
34. 327: what is meant with "rarefied S. aureus abundance"?
35. 332: please explain how the Beta diversities differ.
36. Discussion, lines 342-343: should be outlined already in the Results section.
37. 358: The fact that some previously reported bacterial species were not found in this study may reflect the use of the species database. The eHOMD database, for example, lacks certain species. The authors may consider including also other databases.
38. 363: "negative association of S. aureus with..."
39. 371: "mutualism is a too strong term. The study just shows co-occurrence, not even correlation.
40. Line 382: replace "risk" e.g. with "likelihood".
41. 388-390: Or other colonizing species may shape in vivo interactions between S. aureus and C. accolens (high-order interactions). Studying associations between S. aureus and a group of species (instead of one) may be useful.
42. Line 393-386: nasal S. epidermidis isolates have also been found to produce S. aureus-inhibiting antimicrobial molecules.
43. 405: S. aureus dominates microbiomes only in ca. 50% of the cases.
44. 425-431: sounds strange, see point 7 and 25 above.
45. 436-438: sounds strange, see points 9 and 24 above
46. 438: Or from other body sites (gut for instance)?
47. All figures: characters are often much too small and can be read in printouts only with a magnification glass.
48. Fig 3 A-D: This figure and the associated results/legends are difficult to understand, please clarify. The x axis label is missing on the Panel F3 .

Reviewer #2

(Remarks to the Author)

The paper by Aggarwal et al describes the largest study to date examining the composition of the nasal microbiome in the context of culture-positivity for Staph. aureus. The work provides new insights and clarifications regarding the colonisation status for the studied human population in England. Overall, the study is well written and clearly presented and represents a useful contribution to the field. Overall, the work appears to be robustly carried out with appropriate analysis and well supported conclusions. have a number of comments and queries for the authors.

The findings regarding the nature of intermittent carriage and their association with the composition of either persistent or non-carriers are convincing and an important validation of previous proposals for revision of status from van Belkum et al.

In recent years, it has become established that S. aureus can colonise the human gut and that this may affect human nasal carriage rates. This work should be cited and its relevance for the current study (if any) discussed.

The small but significant number of culture-negative samples that had a high abundance of S. aureus reads is not really commented on- do the authors think this is just down to technical issues regarding self sampling or are there other issue that could culture-negativity ? Did these individuals have consistently negative cultures over the 3 weekly swabs ?

It is unclear from the composition plots what proportion of samples did not result in any *S. aureus* sequence reads? i.e. Are there any truly *S. aureus*-negative samples or do *S. aureus*-culture negative individuals really just have low abundance *S. aureus* which are kept in check by the CST they exist in?

The application of machine-learning for making predictions of colonisation status are as predicted overall but I'm left unsure of the clinical utility for this approach. From what the authors infer, it is *S. aureus* nasal abundance that is the likely risk factor for hospital infection. In which case, a simple QRT-PCR or direct sequencing analysis would provide a similar or even more robust indicator of the risk? It doesn't identify individuals with a higher risk for persistent colonisation as they are presumably already persistently colonised when tested?

I'm not sure the suggested lineage-dependency effect is very convincing. Though I'm sure the stats are OK, it's a relatively small sample size and not sure the power is powered to test this hypothesis. Essentially the species is split into 2 subpopulations A and B which will each comprise many divergent lineages which in turn contain strains with highly variable accessory genomes. Any effect is likely to be influenced by strain-dependent genes e.g. those on MGE as mentioned in the Discussion. This highlights a limitation of the 16 srRNA approach. A shotgun sequencing approach with metagenome assembled genomes may have facilitated the identification of genetic elements associated with persistent colonisation. I think this is worth discussing.

The authors have identified some interesting positive and negative correlations in the samples that may influence the microbiome profile and will be interesting to explore further.

Version 2:

Reviewer comments:

Reviewer #1

(Remarks to the Author)

The authors addressed all my concerns and improved the manuscript significantly. I have only a few remaining minor points:

1. The title still seems a bit confusing. It is not the microbiome that redefines *S. aureus* colonization but its large-scale analysis. How about something like:

"Large-scale characterization of the nasal microbiome redefines *Staphylococcus aureus* colonization status"

2. I strongly suggest using other numbers (maybe roman?) for the seven CSTs as some of those defined by the authors are entirely different from the ones published before by Liu et al. Considering how often the Liu study has formed the basis of other studies, it would evoke a lot of confusion if this study would use the same numbers for a different set of CSTs.

3. It is interesting to note that the PERMDISP test is also significant, meaning that the tested groups have different variances on top of having different means (as tested by the PERMANOVA). These different variances can actually bias the result of the PERMANOVA test.

I would advise to mitigate a bit the following sentence in the result part, especially about the NMDS since the clustering of the different groups is not really clear on Fig. 1F on top of having a significant PERMDISP. Lines 154-159: "Distinct separation of samples by colonisation status could be clearly observed by non-metric multidimensional scaling (NMDS) (Fig. 1F) and PCoA (Fig. 1G) ordination plots. However, samples from intermittent carriers did not form a distinct cluster, and instead overlapped within the persistent or the non-carrier clusters, but with more samples from intermittent carriers being clustered with the non-carriers as visualised by the overlap in data ellipses in Fig. 1F, G"

Also, the last sentence is a bit contradictory to what was said before about a distinct separation of samples by colonisation status. If this is the case, why do samples from intermittent carriers not form a distinct cluster? This needs to be clarified.

4. Fig. 6E: The lower panel is much too small and the resolution too poor.

Reviewer #2

(Remarks to the Author)

The authors have addressed all of the comments to my satisfaction.

Rebuttal for Aggarwal et al

We sincerely thank both reviewers for their detailed and insightful assessment of our manuscript – and in particular for the time they have afforded to provide these evaluations. We greatly appreciate their recognition of the study's novelty, methodological rigor, and contribution to understanding *Staphylococcus aureus* nasal colonization at population scale. The reviewers' thoughtful comments have helped us refine our analyses and improve our interpretations. We have revised the manuscript accordingly to address their concerns and believe the changes enhance the clarity, depth, and impact of our work. Below, we provide detailed point-by-point responses to each comment.

Reviewer 1

Major Comments

1. Nasal microbiomes are analyzed by 16S sequencing, which is often regarded as yielding information only on the genus but not on the species level. Nevertheless, the authors present abundances of species. They obviously amplified other 16S regions (V1-V2) than previous studies, they refer to (V1-V3 or V3-V6). The reasons for their choice and potential relevance for differences to findings in the other studies should be explained in detail in the Results section and they might be addressed in the discussion.

We appreciate the reviewer's concern. While it is true that 16S rRNA sequencing is typically restricted to genus-level classification, prior studies have demonstrated that the V1–V2 region enables improved species-level resolution for key nasal taxa such as *Staphylococcus*, *Corynebacterium*, and *Dolosigranulum*¹⁻³. We now include an expanded explanation in the Methods with associated references clarifying our rationale for selecting this region (Lines 577-582).

2. The authors refer to the 7 community state types previously defined by Liu et al but they obviously use quite a different definition for their 7 CSTs. The CST definition of Liu et al should be described in the introduction and the differences in the present

study, in particular dominance by other bacterial species, should be mentioned in more detail. I also suggest using other names for the CSTs (e.g. A-G instead of 1-7) to avoid confusion. Several rare CSTs were also found and combined in the new CST7. It would be interesting to point out how rare these unusual CSTs are and how they are dominated.

Thank you for pointing this out. The differences with Liu et al are likely to represent differences in the data but not an intrinsic difference in understanding of what CSTs are. The CSTs which we define differ from Liu et al due to a combination of factors; we use of a gap statistic to define the optimal number of clusters and this is visualised on the heatmap. Liu et al in comparison do not detail the method they have used to choose the ideal number of clusters and simply state that “CSTs were identified using a proportional abundance-based matrix in Euclidean distance by hierarchal clustering with Ward linkage”, from which we can only infer that they were chosen visually (Liu et al’s definition of CSTs is now included in the Introduction, lines 89-93). We believe our clustering method and results are more robust; this stems from the use of a gap statistic, a more representative and larger dataset, rarefication of data with greater than 10,000 reads, and meticulous contaminant removal; this is the most likely reason for difference in community state types observed.

We agree that the heterogenous CST7 is particularly interesting, and the detection of diverse sub-clusters is likely to reflect the large sample size of the study. We have provided the new **Supplementary Fig. S8** to explore this cluster further and have commented on this in the Results (Line 206-208) and expanded our commentary on this in the Discussion (Lines 358-363).

3. 16S data are correlated with *S. aureus* culture data but how the abundances correlate is not addressed. What percentage of *S. aureus* culture-negative samples have *S. aureus* 16S rDNA (or the other way around)? Previous studies have suggested that the majority of human nasal microbiomes contain at least some *S. aureus* 16S but that culture positivity depends on a specific bacterial threshold

density. Can such a threshold be defined? This would be highly important for the future use of 16S-based diagnostics.

This is an important suggestion. We now present a comparison between *S. aureus* culture results and 16S rDNA detection. Notably, we observed that although a minority of culture-negative samples contained no *S. aureus*, the majority contain *S. aureus* at an abundance less than 1% (Lines 191-198). We have provided histogram and density plots of *S. aureus* abundance in culture negative and positive samples for readers to explore in the Supplementary materials (**Supplementary Fig. S6**). The focus of this study is to understand the microbial basis of longitudinal *S. aureus* carriage; the random forest model presented (**Fig. 5**) highlights the importance of *S. aureus* abundance as the main predictor of *S. aureus* carriage (longitudinal culture positivity). However, we also show that the additional resident microbial community members contribute to the predictive value of the model and therefore we would lose value in ignoring this data (**Fig 5C and 5D**). We do agree that the practical application of this as a diagnostic need to be evaluated against a *S aureus* qPCR, and we have addressed this point in the discussion (Line 451).

4. The data could be used more extensively to describe positive or negative associations between nasal bacterial species that could instruct detailed functional studies in the future. Considering the size of the dataset, the analysis could go beyond pairs of species, towards combinations of species.

Thank you for this insightful comment; we agree that our large dataset can be further utilised to provide insights on species associations that could be further explored for complex inter-species relationships. We have therefore undertaken significant additional analysis by performing a network-based correlation analyses to capture associations beyond pairwise comparisons and presented this in Results (Lines 228-237), in **Fig. 3B**, and commented on in the Discussion (Lines 383-389).

5. Previous studies have demonstrated antagonistic interactions of *S. aureus* with species such as *Fingoldia magna*, *Staphylococcus lugdunensis*, and others. Do the authors find similar associations? Even if they cannot confirm these data it would be valuable to address the point.

We thank the reviewer for highlighting this. We re-examined associations with previously implicated antagonists such as *Fingoldia magna* and *Staphylococcus lugdunensis*. Both organisms were identified within the dataset but filtered out due to strict contaminant removal criteria (species with an abundance <0.1% across all samples). This reflects thresholds determined in previous work, below which species were not reliably identifiable or distinguishable from contaminants⁴. Further, although it is plausible that species with such a low abundance could have a biologically significant role –it is unlikely and not feasible to determine the relevance of these compared to possible contaminants without the systematic use of technical and biological replicates. Of course, our findings may also reflect true differences in the community compositions between study cohorts or issues with resolution of 16S sequencing. We now explicitly acknowledge these findings in the Discussion (Lines 365-373).

Minor Comments

6. Reconsider the title. It is known that the nasal microbiome defines the capacity of *S. aureus* to colonize but how does it redefine colonization?

We thank Reviewer 1 for making this point. Here, we argue that previous definitions for *S. aureus* colonisation (“non-”, “intermittent-”, and “persistent-” carriage) do not best reflect the relationship of *S. aureus* with colonisation of its human host, i.e. two key states: a *S. aureus* dominated CST in which *S. aureus* shapes the microbiome, and a group of CSTs in which *S. aureus* is rare or absent. As we discuss in our manuscript (Lines 66-72), this is supported by previous immunological studies. We also offer the most parsimonious explanation for intermittent carriage, who, given their low *S. aureus* abundance and predominance amongst non-carriage CSTs, are

effectively *S. aureus* 'non-dominant carriers' who are only transiently colonised with *S. aureus*, for example due to environmental/household (Lines 464-471).

7. Abstract, lines 57-58: defining the *S. aureus* colonization statuses should require only the detection of *S. aureus* 16S DNA but machine learning could help to predict its stability (persistent vs. intermittent) right?

Thank you for requesting this clarification, we have accordingly edited the abstract to highlight machine learning aids in predicting the *persistence* of colonisation.

8. Line 59: "likely better adapted" than what?

Thank you for highlighting this error in wording, now amended to specify the comparator.

9. Results, lines 74-79: the logic in this sentence is not fully clear. How do all these points support the notion that there are only two categories? Moreover, the sentence implies that intermittent carriers do not exist but the authors probably mean that intermittent carriers are not a distinct community state type.

Thank you for making this important point and we have now clarified this point. The lack of an intermittent carriage related microbiome has not been shown before, and a key finding from our study. However, there is evidence to suggest intermittent and non-carriers have similar colonisation dynamics, and are distinct from persistent carriers which suggests the existence of two, rather than three, *biologically relevant* phenotypes. This has now been reflected in the introduction (Lines 65-72).

10. Line 81: please specify "colonization", colonization rate or *S. aureus* nasal abundance?

We now specify “colonisation prevalence” to avoid ambiguity.

11. Line 82: better “...though many of these studies are based on only a small number of study participants...”

Rephrased to: “Though many previous studies are based on small numbers of participants...” (Lines 74-75).

12. Line 91: “lower overall diversity” compared to?

Specified the comparison group for “lower diversity” as the middle meatus and sphenoethmoidal recess, and non-persistent carriers (Lines 83-84).

13. Line 97: add “sp” or “spp” to genus names

Added “sp.” to appropriate genus names for clarity.

14. Line 98: “lower overall bacterial density” compared to? “16S rRNA gene”

We have stated that this is amongst women and therefore the comparator to men is implied. We have however clarified the measurement by stating “lower overall bacterial density as measured by 16S rRNA gene copy number.” (Line 92-93).

15. Line 106: What is meant with “positive integration”?

Rephrased “positive integration” to “co-occurrence within the broader microbial community.” (Lines 100-101).

16. Line 154: define QC

Defined "QC" as "quality control".

17. 159-160: Remove one „S. aureus“

Thank you, removed duplicated "S. aureus."

18. Lines 165-157: sounds redundant with the sentence before

We agree that the *S aureus* culture based alpha diversity differences do not need to be explicitly stated given the previous sentence and have deleted accordingly.

19. 170-171: Regarding the variance within their groups, better also add the PERMDISP statistical results, at least in the figure legends.

PERMDISP results have been calculated and are now provided in the legend of **Fig. 1**. Additionally, details of the test have been added to the methods section (Lines 680-683).

20. 172-173: wrong figure cited (1G)

Thank you for noting this, erroneous figure citation has been corrected.

21. Lines 179- 181: sounds weird – S. aureus culture positivity should be associated with both, persistent and intermittent carriage

We agree that this does sound odd and we have deleted the comparison. Persistent carriers are, by definition, *S. aureus* culture positive in all three nasal swabs and

therefore the association is strongest with this category. Intermittent 'carriers' are dispersed across NMDS and PCoA plots, and we later show that they form two distinct clusters based on the number of swabs positive – the association is therefore not as clear cut as with persistent carriers. We do, however, agree that the sentence introduces confusion.

22. Lines 186-187: sounds contradictory with the sentence before.

Thank you for noting this error, we have now edited the text to correctly reflect the results.

23. Lines 194-195: make clear that this statement refers to a new CST classification that differs considerably from the previous CST scheme. Explain how the new CSTs were defined (like which threshold for assuming different CSTs).

We have detailed the subtle differences in CST classification when compared to previous work in the Introduction and Discussion, we have therefore not made further clarification in the Results section. We do however detail how we chose the optimal number of clusters and then applied this to the dendrogram ordered by hierarchical clustering of bray-curtis distances (Lines 204-207)

24. 203: "pseudodiphteriticum"

Corrected spelling to "pseudodiphtheriticum".

25. Lines 207- 210: sounds again unclear and potentially contradictory to the statements in the paragraph before.

We have now added detail to this work which provides greater clarity to the results. We hypothesised that an important association of *S. aureus* culture positivity with male sex could be, in part, explained by biases in culture as we observed a small proportion of samples with a high *S. aureus* abundance but from which we did not culture *S. aureus*; we report our findings that there was not an appreciable difference in samples with a high *S. aureus* abundance but which failed to culture *S. aureus* between the sexes (Lines 191-198).

26. Line 230: abundance of what?

We have edited these results to provide clarity (Lines 216-218).

27. Line 232-233: “positive association of *S. aureus* with persistent carriage” sounds trivial.

Thank you, although this may appear a trivial finding, in Lines 216-227 we draw out the significance when comparing the extent of the difference in abundance of *S. aureus* amongst persistent carriers with non-carriers, when compared with intermittent carriage. This difference also suggests that *S. aureus* plays a self-fulfilling role to occupy the anterior nares over time and possibly exclude competitive organisms (discussed in Lines 421-433).

28. Line 245: what is meant with “pairwise Alpha diversity”?

Clarified that “pairwise alpha diversity” refers to the comparison of diversity indices from samples of the same participant between consecutive weeks of sampling (Lines 241-243).

29. Line 251: "...the majority of microbiomes of intermittent carriers clustered with those of..."

Thank you for providing this suggested edit – it improves the sentence structure and we have incorporated it in the text (Lines 248-257).

30. 263: Wrong figure cited (3E)

Thank you for pointing out this error – now edited in the text and now **Fig. 4E**.

31. 280: please rephrase, refer to point 9 above.

Thank you, we have clarified the text to specifically state that intermittent carriers do not possess a unique microbiome.

32. Three paragraphs starting at line 183: please address how the abundance of *S. aureus* 16S compares to culture positivity. To my understanding, overall microbiome composition should help to predict the stability of *S. aureus* colonization while the presence of *S. aureus* should be deduced from the detection of its rDNA (see my point 7)? But then the authors state in line 300 – 301 that their model was unable to predict intermittent colonization. Please explain better, I am confused...

Thank you for raising this point. In the section, "Predicting colonisation status from the nasal microbiome", we examine the accuracy of the normalised nasal microbiome data (derived from a single swab) to predict the colonisation state of an individual, which would otherwise be derived from three consecutive culture-positive swabs. As Reviewer 1 correctly states, *S. aureus* abundance is certainly the most important determinant of the model, which is demonstrated in **Fig. 5C** (Mean Decrease in Gini Index) and **5D** (Mean Decrease in Accuracy). However, *S. aureus* abundance is not the only determinant; we show that multiple other species such as,

Corynebacterium sp., and *S. epidermidis*, and *D. pigrum*, are biological markers of the colonisation states.

As Reviewer 1 indicates, the microbiome does predict colonisation states (and therefore stability to some extent) with good accuracy. The model performs well in predicting non- or persistent carriage but *does not* predict intermittent carriage. The reason for this, as we allude to throughout the text, is that intermittent carriers do not have a distinct microbiome (and as a circular argument, the microbiome does not predict intermittent carriage). We interrogate the output of the model further and provide clear evidence that the Random Forest model incorrectly classifies intermittent carriers as *either* non-carriers or persistent carriers. In the discussion, we provide the most parsimonious explanation for this (Lines 464-471).

We are *not* predicting *S. aureus* culture positivity with *S. aureus* abundance data. with this Random Forest model.

We separately also graphically present the relationship of *S aureus* abundance with *S aureus* culture for readers to explore as requested in Point 3 (**Supplementary Fig. S6**).

33. Line 323-329: difficult to understand, please rephrase. How are the two clusters defined? Mention also the dominating STs in cluster A. The whole paragraph would profit from a clear conclusion. It would be helpful also to state how the identified nasal STs compare to the abundant invasive STs in the UK.

Thank you for requesting this clarification. We now provide a definition for the two major *S. aureus* clusters, describe their dominant STs, and have added a clear conclusion (Lines 318-323). For readers to be able to contextualise the nasal sequence-types, we have provided information on the invasive *S aureus* isolates in the Discussion (Lines 474-475).

34. 327: what is meant with “rarefied *S. aureus* abundance”?

Clarified that “rarefied abundance” refers to per-sample read normalization.

35. 332: please explain how the Beta diversities differ.

Thank you for this comment. Possible differences in distribution of beta diversity have now been reported with a PERMDISP analysis (**Fig. 6** legend). Compositional differences between the clusters are represented in the heatmap accompanying **Fig. 6A**.

36. Discussion, lines 342-343: should be outlined already in the Results section.

Thank you for pointing this out. We have kept the results pertaining to this study in the Results section but as per Reviewer 1’s suggestion, we have highlighted differences with Liu et al in the Introduction and the Discussion sections. Lines 342-343 have been retained as they are an important interpretation of the data, and a basis for future work.

37. 358: The fact that some previously reported bacterial species were not found in this study may reflect the use of the species database. The eHOMD database, for example, lacks certain species. The authors may consider including also other databases.

Please see our response to point 5. Some previously reported bacterial species were in such low abundance that they are automatically filtered out in our pipeline, which is stringent for contamination removal based on previous validity work⁴.

We agree that taxonomic assignment with different databases can result in some differences in species. However, we use a highly cited database, Silva bacterial database ‘silva.nr_v132.align’, which has been employed in multiple high-quality studies⁵⁻⁷. Utilising more than one database for taxonomic assignment is not routine practice; we have gone to additional lengths to reduce uncertainty in our data by

manually searching taxa for incorrectly assigned species with GenBank (see **Supplementary materials**). Further, our sequence data has been made available for readers to replicate the work, we provide the taxonomic assignment of 'OTUs' in supplementary data, and to more robustly address the point made by Reviewer 1, we now make the raw unique 'OTU' fasta file available for readers to rapidly cross-check taxonomic assignment (DOI: 10.5281/zenodo.17160107). Finally, although re-assigning taxa with various databases is beyond the scope of this study, we highlight this important point as a limitation of microbiome studies (Line 493).

38. 363: “negative association of *S. aureus* with...”

We thank Reviewer 1 for highlighting this grammatical error – now edited.

39. 371: “mutualism is a too strong term. The study just shows co-occurrence, not even correlation.

Thank you, we agree that we show co-occurrence rather than “mutualism”. We have now replaced “mutualism” with “co-occurrence” to avoid overstating the nature of the relationship (Line 382). The suggested network analysis (Point 4) now captures patterns of association based on statistical co-dependencies between taxa, reflecting potential correlation structures (Lines 381-389).

40. Line 382: replace “risk” e.g. with “likelihood”.

Thank you for pointing out this error - replaced “risk” with “likelihood”.

41. 388-390: Or other colonizing species may shape in vivo interactions between *S. aureus* and *C. accolens* (high-order interactions). Studying associations between *S. aureus* and a group of species (instead of one) may be useful.

This is a valid and interesting hypothesis; we have expanded the discussion to acknowledge possible higher-order interactions with groups of species and the need for future studies to consider these (Lines 407-409).

42. Line 393-386: nasal *S. epidermidis* isolates have also been found to produce *S. aureus*-inhibiting antimicrobial molecules.

Thank you for this comment – we have now added a statement regarding antimicrobial-producing nasal *S. epidermidis* isolates with the appropriate citation (Line 410-420).

43. 405: *S. aureus* dominates microbiomes only in ca. 50% of the cases.

Thank you – we agree that our statement overstated the presence of *S. aureus* and we have removed the final part of this sentence.

44. 425-431: sounds strange, see point 7 and 25 above.

Thank you, please see our response to point 25 above. *S. aureus* colonisation states have been recognised as biologically and clinically important phenotypes – we used the microbiome data to examine the ability to predict *S. aureus* carriage from one swab. This model was effective with good accuracy, but notably it's reduced accuracy is due to the lack of a microbiome signature for intermittent carriage. The model recognised the microbiomes of intermittent carriers as *either* non- or persistent- carriers. Despite this, the model performs well in its ability to *rule out* non-persistent states, which we argue is of clinical importance.

45. 436-438: sounds strange, see points 9 and 24 above

We have again provided clarity to the sentence (Lines 464-468). Importantly, although 'intermittent' carriage may be a phenotype defined by culture, our data on the nasal microbiome and previous work examining experimental colonisation, abundance, and immunological response to colonisation, appear to suggest these individuals appear have biomarkers that are either like non-carriers or persistent carriers. We therefore provide the most parsimonious explanation of this:

“‘intermittent’ carriers, given their low S. aureus abundance and predominance amongst non-carriage CSTs, are effectively S. aureus ‘non-dominant carriers’ who are only transiently colonised with S. aureus, for example due to environmental/household exposure (i.e. hypothesis 2 proposed by Van Belkum et al.). While two swab positive ‘intermittent’ carriers are ‘S. aureus dominant carriers’ (akin to persistent carriers), reflected by their higher S. aureus abundance and greater representation amongst the S. aureus dominant CST, and therefore likely to be persistently colonised individuals that were negative by culture in one swab.”.

This work alone provides an important finding – that the intermittent carriers do not have a distinct nasal microbiome. It also supports work which suggests that intermittent carriers are not a distinct biological phenotype through immunological and experimental inoculation data; together these lay the foundations for designing future studies and better understanding host-*S. aureus* relationships.

46. 438: Or from other body sites (gut for instance)?

Thank you, this is an important point. We have added consideration that recolonization may also originate from other body sites, such as the gut (Line 501-504).

47. All figures: characters are often much too small and can be read in printouts only with a magnification glass.

Figure fonts have now been enlarged for readability. We have also split **Fig. 2** in order to improve the clarity of text. Please note, **Fig. 6E** is intended to be a graphical representation of the abstract and the axis labels of the miniaturised figures not relevant to it.

48. Fig 3 A-D: This figure and the associated results/legends are difficult to understand, please clarify. The x axis label is missing on the Panel F3 .

Thank you, we have now updated the figure legends for clarity (now **Fig. 4**), corrected axis labelling, and improved annotation of panel F. Specifically, to clarify the figure and its legend, **Fig. 4** explores the microbial community structure in the anterior nares in relation to *Staphylococcus aureus* colonisation status, but with a particular emphasis on intermittent carriers. The figure comprises four ordination plots based on Bray-Curtis dissimilarity, visualised using two methods: NMDS (panels a, c) and PCoA (panels b, d). Panels **4A** and **4B** reproduce ordination plots from **Fig. 1F** and **1G**, but here the intermittent carriers (blue) are visually emphasised, while persistent carriers and non-carriers are faded. This highlights that intermittent carriers occupy a transitional position in microbial community space, spanning the clusters associated with both persistent carriers and non-carriers.

Panels **4C** and **4D** present the same ordination space as panels **4A** and **4B**, but points are recoloured by the number of *S. aureus*-positive swabs per participant (0 to 3). This gradient-based colouring reveals a continuum: individuals with 1 positive swab tend to cluster near non-carriers (0 swabs), while those with 2 positive swabs align more closely with persistent carriers (3 swabs). Together, these plots suggest that intermittent carriers are not a discrete group but rather span a gradient of microbial community composition.

Reviewer 2

1. In recent years, it has become established that *S. aureus* can colonise the human gut and that this may affect human nasal carriage rates. This work should be cited and its relevance for the current study (if any) discussed.

We thank the reviewer for raising this important and timely point. In the limitations section of the Discussion, we have now highlighted *S. aureus* colonization of the human gut and cited relevant studies. We specifically comment on the need to systematically examine within-host cross-niche transmission of *S. aureus* and re-examine the persistence of colonisation across multiple host niches in large and generalisable cohorts. While we did not collect gastrointestinal samples in this study, we agree this represents a compelling direction for future investigations (Line 504-507).

2. The small but significant number of culture-negative samples that had a high abundance of *S. aureus* reads is not really commented on- do the authors think this is just down to technical issues regarding self sampling or are there other issue that could culture-negativity ? Did these individuals have consistently negative cultures over the 3 weekly swabs?

This is an excellent observation. We now explicitly address this discrepancy in the Discussion (Lines 507-511). Several factors may explain the detection of high *S. aureus* 16S abundance in culture-negative samples: (a) unfavourable transport/storage conditions for a small number of samples, (b) non-culturability of the *S. aureus* strain due to auxotrophism or being in a viable but nonculturable state, or (c) the presence of dead bacteria at the time of sampling due to exposure to antimicrobials or the action of the immune system. We also examined whether these individuals were consistently culture-negative across three timepoints and found that in most cases, the high 16S abundance was transient, suggesting intermittent carriage or sampling variability (**Supplementary Fig. S6**).

3. It is unclear from the composition plots what proportion of samples did not result in any *S. aureus* sequence reads ? ie Are there any truly *S. aureus*-negative samples or do *S. aureus*-culture negative individuals really just have low abundance *S. aureus* which are kept in check by the CST they exist in?

We appreciate this important question. We have now added additional data in the Results section (Lines 191-198) and Supplementary Materials (**Supplementary Fig. S6**) clarifying the distribution of *S. aureus* 16S abundance across all samples. A subset of individuals had no detectable *S. aureus* reads across all swabs and were considered truly negative. However, the majority of culture-negative individuals had detectable but very low-abundance *S. aureus*, supporting the idea that CST composition may play a role in keeping *S. aureus* subdominant or alternatively this represents exposure of the individuals to *S. aureus* followed by the death of *S. aureus* due to interaction with other microbiome members, nasal environment or the immune response.

4. The application of machine-learning for making predictions of colonisation status are as predicted overall but I'm left unsure of the clinical utility for this approach. From what the authors infer, it is *S. aureus* nasal abundance that is the likely risk factor for hospital infection. In which case, a simple QRT-PCR or direct sequencing analysis would provide a similar or even more robust indicator of the risk? It doesn't identify individuals with a higher risk for persistent colonisation as they are presumably already persistently colonised when tested?

We agree with the reviewer that this is a key consideration. The Discussion now acknowledges that qPCR of *S. aureus* abundance may represent an alternate to microbiome stratified clinical risk inference for infection or transmission (Line 450-452). Our machine-learning approach demonstrates proof-of-principle that nasal community composition encodes predictive information - our model was not designed to predict *future* colonization but to uncover features associated with carriage stability at the time of sampling. Persistent colonisers are known to be at greater risk of *S. aureus* infections⁸; the ability to accurately predict this phenotype from one swab certainly presents possible utility. Reviewer 2 is correct that *S. aureus* abundance is certainly the most important determinant of the model, which is demonstrated in **Fig. 5C** and **5D**. However, *S. aureus* abundance is not the only determinant; we show that multiple other species such as, *Corynebacterium sp.*, and *S. epidermidis*, and *D. pigrum* are biological markers of the colonisation states.

Importantly our model also has a high negative-predictive value, which again provides important information to risk-stratify patient pre-operatively, for example.

5. I'm not sure the suggested lineage-dependency effect is very convincing. Though I'm sure the stats are OK, its a relatively small sample size and not sure the power is powered to test this hypothesis. Essentially the species is split into 2 subpopulations A and B which will each comprise many divergent lineages which in turn contain strains with highly variable accessory genomes. Any effect is likely to be influenced by strain-dependent genes eg those on MGE as mentioned in the Discussion. This highlights a limitation of the 16 srRNA approach. A shotgun sequencing approach with metagenome assmbled genomes may have facilitated the identification of genetic elements associated with persistent colonisation. I think this is worth discussing.

We thank the reviewer for this nuanced critique. We acknowledge that our sample size limits the statistical power to make definitive claims about lineage-level associations. In response, we have discussed the need for this work to be assessed in a larger cohort of individuals. We also discuss the need for the potentially confounding impact of mobile genetic elements on the relationship of *S. aureus* lineages and the resident microbes to assessed with a shotgun metagenomic approach (Lines 482-485).

6. The authors have identified some interesting positive and negative correlations in the samples that may influence the microbiome profile and will be interesting to explore further.

We appreciate the reviewer's interest in these association and we have commented on the utility of these findings for future studies and their potential implications for microbial interactions, colonisation resistance, and therapeutic targeting in Lines 349-351 of the Discussion. We now additionally provide a network analysis that provides analysis beyond pairwise comparisons of species with further potential targets for 'community-based' co-culture studies to understand the possible co-dependent relationships in excluding or supporting the growth of *S. aureus* (Lines 388-389).

References

- 1 Lopez-Aladid, R. *et al.* Determining the most accurate 16S rRNA hypervariable region for taxonomic identification from respiratory samples. *Sci Rep* **13**, 3974, doi:10.1038/s41598-023-30764-z (2023).
- 2 Chakravorty, S., Helb, D., Burday, M., Connell, N. & Alland, D. A detailed analysis of 16S ribosomal RNA gene segments for the diagnosis of pathogenic bacteria. *J Microbiol Methods* **69**, 330-339, doi:10.1016/j.mimet.2007.02.005 (2007).
- 3 Escapa, I. F. *et al.* New Insights into Human Nostril Microbiome from the Expanded Human Oral Microbiome Database (eHOMD): a Resource for the Microbiome of the Human Aerodigestive Tract. *mSystems* **3**, doi:10.1128/mSystems.00187-18 (2018).
- 4 Aggarwal, D. *et al.* Optimization of high-throughput 16S rRNA gene amplicon sequencing: an assessment of PCR pooling, mastermix use and contamination. *Microb Genom* **9**, doi:10.1099/mgen.0.001115 (2023).
- 5 de Goffau, M. C. *et al.* Gut microbiomes from Gambian infants reveal the development of a non-industrialized Prevotella-based trophic network. *Nat Microbiol* **7**, 132-144, doi:10.1038/s41564-021-01023-6 (2022).
- 6 Wagner, J. *et al.* The composition and functional protein subsystems of the human nasal microbiome in granulomatosis with polyangiitis: a pilot study. *Microbiome* **7**, 137, doi:10.1186/s40168-019-0753-z (2019).
- 7 Yilmaz, P. *et al.* The SILVA and "All-species Living Tree Project (LTP)" taxonomic frameworks. *Nucleic Acids Res* **42**, D643-648, doi:10.1093/nar/gkt1209 (2014).
- 8 Nouwen, J. L., Fieren, M. W., Snijders, S., Verbrugh, H. A. & van Belkum, A. Persistent (not intermittent) nasal carriage of *Staphylococcus aureus* is the determinant of CPD-related infections. *Kidney Int* **67**, 1084-1092, doi:10.1111/j.1523-1755.2005.00174.x (2005).

We sincerely thank both reviewers for the time they have afforded to provide these evaluations - we agree that their insights have much improved the manuscript and for this we are grateful. Below, we provide detailed point-by-point responses to each comment.

Best wishes,

Dr Dinesh Aggarwal

Dr Ewan Harrison

(on behalf of the authors)

Reviewer 1

Minor Comments

1. The title still seems a bit confusing. It is not the microbiome that redefines *S. aureus* colonization but its large-scale analysis. How about something like: "Large-scale characterization of the nasal microbiome redefines *Staphylococcus aureus* colonization status".

We appreciate the reviewer's comment and have accordingly changed the title to, "Large-scale characterisation of the nasal microbiome redefines *Staphylococcus aureus* colonisation status".

2. I strongly suggest using other numbers (maybe roman?) for the seven CSTs as some of those defined by the authors are entirely different from the ones published before by Liu et al. Considering how often the Liu study has formed the basis of other studies, it would evoke a lot of confusion if this study would use the same numbers for a different set of CSTs.

We thank Reviewer 1 for this point. Although we have used numbering that is convention for microbiome studies and a changing CST profile will reflect different

methods and study cohorts (as discussed in our previous response relating to this point), we appreciate the reviewer's concern of the impact on the wider scientific community and have therefore used roman numerals, as suggested.

3. It is interesting to note that the PERMDISP test is also significant, meaning that the tested groups have different variances on top of having different means (as tested by the PERMANOVA). These different variances can actually bias the result of the PERMANOVA test.

I would advise to mitigate a bit the following sentence in the result part, especially about the NMDS since the clustering of the different groups is not really clear on Fig. 1F on top of having a significant PERMDISP. Lines 154-159 : "Distinct separation of samples by colonisation status could be clearly observed by non-metric multidimensional scaling (NMDS) (Fig. 1F) and PCoA (Fig. 1G) ordination plots. However, samples from intermittent carriers did not form a distinct cluster, and instead overlapped within the persistent or the non-carrier clusters, but with more samples from intermittent carriers being clustered with the non-carriers as visualised by the overlap in data ellipses in Fig. 1F, G"

Also, the last sentence is a bit contradictory to what was said before about a distinct separation of samples by colonisation status. If this is the case, why do samples from intermittent carriers not form a distinct cluster? This needs to be clarified.

We thank Reviewer 1 for this comment. Although we agree that the significant PERMDISP result can bias the PERMANOVA result, the variance in dispersion should be interpreted in the context of the ordination plots. We clearly see separation between persistent and non-carriers - who also have an observed difference in dispersion. That said, we agree adding the PERMDISP result and providing a level of mitigation adds value to the statistical rigor of the work and we have now added this (Lines 185-195). We have also adjusted the statement, "distinct separation of samples by colonisation status" to indicate this represents separation of persistent- and non-carriers.

4. Fig. 6E: The lower panel is much too small and the resolution too poor.

We thank Reviewer 1 for this observation. Figure 6E is a cartoon illustration and reflects the conclusions of the study from the analyses conducted. We have provided a high resolution image to the journal and a link to a Biorender hosted image which allows readers to view the original graphic and expand as they wish (<https://BioRender.com/o73y544>).

Reviewer 2

The authors have addressed all of the comments to my satisfaction.

Thank you, we are pleased to have satisfactorily addressed Reviewer 2's comments.